# CortexVLA: Bridging the Gap between Cognition and Action via Function Calling

## Abstract

Vision-Language-Action (VLA) models have shown promise for embodied intelligence, but they often struggle with long-horizon tasks due to error accumulation or planning failure. To address these challenges, we propose CortexVLA, a novel paradigm that bridges cognition and action by leveraging large language model (LLM) function calling. CortexVLA consists of three modular components: the **Central Cortex**, an LLM-based cognitive hub for planning and function calling; the **Visual Cortex**, which provides perception through callable vision tools; and the **Motor Cortex**, which exposes robotic action control as functions. To improve robustness and enable recovery from execution errors, we further propose **Cortex-PPO**, a reinforcement learning (RL) algorithm that trains CortexVLA to make optimal function calls while supporting failure recovery. We provide theoretical analyses to further demonstrate the soundness and generalization abilities of Cortex-PPO. Comprehensive experiments demonstrate the effectiveness of CortexVLA on ultra-long-horizon tasks. In our main experiment, CortexVLA achieves an average success rate of 85.40%. More importantly, it sustains a 72.73% success rate with an average sub-task length of 11.55 when tackling the most challenging 14 sub-tasks, whereas end-to-end VLA baselines fail beyond 3 or 4 steps. In a flexible manufacturing scenario with 31 sub-tasks, CortexVLA achieves an 81.25% success rate with an average sub-task length of 26.69, demonstrating strong scalability and adaptability. Codes will be released after publication.

## 1 Introduction

Recent advances in vision-language-action (VLA) models (Zitkovich et al., 2023; Wen et al., 2025b) have driven substantial progress in embodied intelligence. By leveraging the powerful visual-linguistic representations learned by pretrained vision-language models (VLMs), these generalist robot manipulation policies are trained to map visual observations and natural language instructions directly into robotic actions (O'Neill et al., 2024; Zhao et al., 2025a; Liu et al., 2025; Zhou et al., 2025). Although most VLAs perform well on short tasks after fine-tuning, they often struggle with long-horizon tasks, frequently leading to incomplete execution or failure (Fan et al., 2025).

The primary reason existing methods struggle with long-horizon tasks is that many approaches (Zhao et al., 2023; Octo Model Team et al., 2024; Kim et al., 2024) operate strictly under the Markov assumption. When the system state deviates from the training distribution, which frequently occurs during sequential execution, the models decisions often lead to failure. To mitigate this limitation, $\pi_0$ (Black et al., 2024) leverages the high-level policy model SayCan (Ahn et al., 2022) to decompose long-horizon tasks into sub-goals. Other works (Wen et al., 2025a;c) exploit the models inherent capabilities, employing internal reasoning or phase-aware adaptation to guide execution. However, these end-to-end methods suffer from error accumulation, with small deviations compounding over long horizons into task failure. In addition, such models typically require fine-tuning on large collections of continuous sequences, which not only raises the cost of data acquisition but also makes performance highly sensitive to data quality, leading to significant instability and difficulty in reproduction. From another perspective, recent hierarchical VLA architectures (Zhang et al., 2025; Shi et al., 2025; Gao et al., 2025) have shown improvement on long-horizon tasks. However, they remain susceptible to errors during planning and task state tracking, and their reliance on synthetic or simulated data for training makes them difficult to deploy in practice. Related works are discussed in Appendix A.

Meanwhile, research on large language models (LLMs) has shown strong capabilities in tool usage (Yao et al., 2023). Several studies (Qin et al., 2023; Liu et al., 2024b; Qian et al., 2025) improve function calling through better data construction and fine-tuning, enabling open-source models such as Llama3 (Dubey et al., 2024) and Qwen3 (Yang et al., 2025) to connect with diverse APIs across a wide range of scenarios. In robotic manipulation, however, high-level cognition of human instructions and environmental context often fails to translate into reliable action control. This naturally raises the question: *Can we harness the cognitive strengths and function-calling abilities of LLMs to better bridge the gap between cognition and action in robotic manipulation?*

To answer this question, a straightforward strategy is to decouple visual perception and action execution into distinct tool modules and let an LLM orchestrate their invocation. However, this design faces critical challenges in long-horizon tasks. As the number of sub-tasks grows, the context length expands proportionally, making it difficult for the model to distinguish tasks that have already been finished from pending tasks. It also increases the risk of premature termination due to context limits. Moreover, when errors occur within a sub-task, the model often struggles to locate mistakes in the lengthy context and re-execute the sub-task. These issues highlight the need for a new collaborative paradigm that enables robots to handle long-horizon tasks more effectively.

In this paper, we propose **CortexVLA**, a novel VLA paradigm that leverages LLMs for natural-language understanding and function calling to drive visual perception and action control. Our framework comprises three core components: the **Central Cortex**, the **Visual Cortex**, and the **Motor Cortex**. The Central Cortex serves as the cognitive hub, receiving user instructions, decomposing them into structured task lists, and orchestrating execution through sequential tool calls to the other two Cortices. To address context-length limitations, it incorporates a **task handler** that persistently stores and updates task states, enabling explicit progress tracking and bounded prompt context. The Visual Cortex provides perception by integrating vision modules (e.g., object detectors, depth sensors) exposed as callable tools. Similarly, the Motor Cortex governs robotic motion by invoking action modules such as pose predictors, motion planners, and low-level controllers. The Visual and Motor Cortices share a unified tool library, which enumerates the available functions for operating their respective modules. Both the library and the underlying modules can be flexibly replaced to adapt to different application scenarios.

To further expand the ability of CortexVLA, we propose **Cortex-PPO**, a reinforcement learning (RL) algorithm tailored for CortexVLA. By introducing a recovery-aware reward and noise injection, this algorithm not only improves the function calling accuracy of CortexVLA but also equips it with failure recovery capabilities, substantially enhancing its robustness in executing long-horizon tasks. We further conduct a theoretical analysis of Cortex-PPO, proving the unbiasedness of noise injection and establishing an upper bound on cross-environment performance generalization based on mutual information, which demonstrates its soundness and generalization benefits.

To evaluate CortexVLA, we conducted extensive experiments across diverse scenarios against VLA baselines. On our main benchmark of ultralong-horizon tasks ranging from 1 to 14 sub-tasks, CortexVLA attains an average success rate of 85.40%, outperforming all baselines. In the hardest setting with 14 sub-tasks, it maintains a 72.73% success rate with an average sub-task length of 11.55, whereas most baselines fail once the number of sub-tasks reaches 3 or 4. In a flexible manufacturing scenario with 31 sub-tasks, CortexVLA achieves an 81.25% success rate with an average sub-task length of 26.69. Case studies across multiple domains further illustrate its flexibility and adaptability. Overall, these results indicate high success rates, strong stability, and reproducibility of CortexVLA. Our contributions are outlined as follows:

- We propose **CortexVLA**, a novel VLA paradigm that leverages LLMs for high-level planning and function calling to coordinate vision and action modules. The architecture consists of a **Central Cortex** for planning and function calling, a **Visual Cortex** for perception, and a **Motor Cortex** for control.

- We design **Cortex-PPO**, a RL algorithm that improves the function calling accuracy of CortexVLA and equips CortexVLA with failure recovery ability. We further provide theoretical analyses that demonstrate its soundness and generalization guarantees.

- We evaluate CortexVLA through extensive experiments across diverse scenarios, demonstrating superior performance, adaptability, and robustness on ultralong-horizon tasks compared to previous VLA baselines.

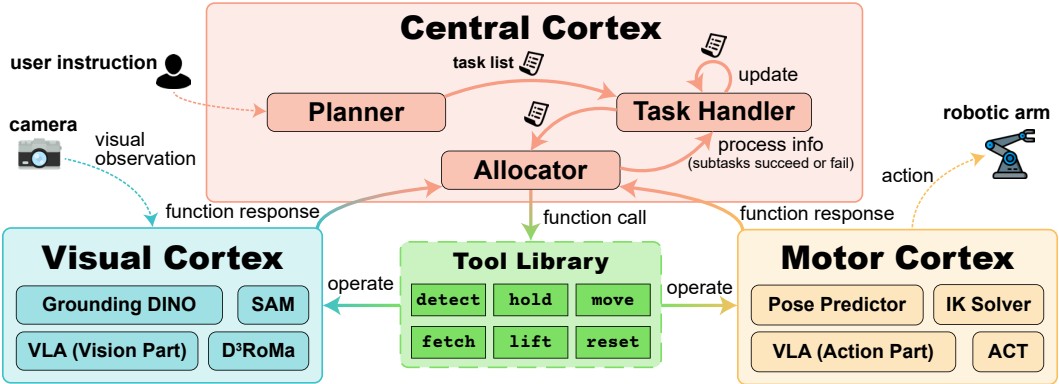

Figure 1: **The overall architecture and basic working flow of CortexVLA.** The functions in the tool library and the modules in the Visual and Motor Cortices can be replaced to adapt to different application scenarios.

## 2 METHODOLOGY

### 2.1 CORTEXVLA

As shown in Figure 1, CortexVLA is composed of three core components: the **Central Cortex**, the **Visual Cortex**, and the **Motor Cortex**. The Central Cortex acts as the control hub, receiving user instructions, maintaining task lists, and orchestrating function calls. The Visual Cortex handles perception, while the Motor Cortex executes actions. Both the Visual Cortex and the Motor Cortex share a tool library that contains the available functions. The tool library, along with the perceptual and motor modules, can be flexibly configured to meet different task requirements. We describe each component in detail below.

**Central Cortex** The Central Cortex comprises two LLM-based decision layers and a task handler. The upper layer is the **Planner**. It receives user instructions, analyzes them, and generates a structured task list that is stored by the task handler. The task list enumerates the sub-tasks, allowing the task handler to track progress and manage updates efficiently. The lower layer is the **Allocator**. It is prompted with the current task list, the current context, and the relevant function descriptions from the tool library. Then, it executes the sequential function calls to engage the Visual Cortex and the Motor Cortex in solving the first sub-task on the task list. Each time after calling a function, it receives a function response that provides critical information or indicates whether the function execution succeeds or fails. If it fails, the Allocator can perform failure recovery according to the recovery strategies defined in the function descriptions, which is essential for long-horizon tasks where errors are unavoidable. Once a sub-task is completed, the task handler updates the task list and provides the updated version to the Allocator. The Allocator can then clear the context without losing essential information. The implementation details are presented in Appendix D.

**Visual Cortex** The Visual Cortex handles all vision-related processing and transmission. It typically connects to external perception devices and incorporates multiple visual modules, which can be used independently or in combination through functions in the tool library. For instance, in our main experiment (Section 3.1), the Visual Cortex is connected to a depth camera and equipped with modules such as Grounding DINO (Liu et al., 2024a), D³RoMa (Wei et al., 2024), and the Segment Anything Model (SAM) (Kirillov et al., 2023). Grounding DINO is invoked independently for object detection, while D³RoMa and SAM are jointly used for point cloud generation. The outputs of the Visual Cortex, including RGB images, bounding boxes, and point clouds, are then passed to the Motor Cortex to guide action execution.

**Motor Cortex** The Motor Cortex is responsible for executing robotic actions by generating action sequences that drive the robotic arm to complete tasks. Like the Visual Cortex, it comprises multiple motor modules that can be invoked independently or in combination through function calls. For instance, in our main experiment (Section 3.1), which involves grasping operations, we em-

ploy AnyGrasp (Fang et al., 2023) as the end-effector pose predictor. The generated actions are further processed by low-level controllers, including inverse kinematics (IK) solvers and obstacle-avoiding motion planners, to produce executable commands. The Motor Cortex then reports execution outcomes back to the Central Cortex (Allocator) through function responses, enabling subsequent decision-making.

The Visual Cortex and Motor Cortex are not always decoupled. When employing models such as ACT (Zhao et al., 2023) or end-to-end VLAs, the visual component (e.g., VLM) naturally plays the role of the Visual Cortex, while the action component (e.g., action expert) serves as the Motor Cortex. For example, in the Bartender case study (Section 3.2.2), we use ACT as both the Visual Cortex and Motor Cortex.

### 2.2 CORTEX-PPO

Equipping CortexVLA with robust scheduling and function call capabilities requires fine-tuning with specific data. A common approach is to perform supervised fine-tuning (SFT) using Low-Rank Adaptation (LoRA) (Hu et al., 2022). While SFT enables the model to familiarize itself with the current scenario and achieve reasonably good function-calling accuracy, it struggles to execute failure recovery operations effectively when errors occur during sub-tasks. To address this issue, we propose **Cortex-PPO**, an RL algorithm for fine-tuning the CortexVLA to operate correctly in failure recovery and perform better function calls. Cortex-PPO also enables CortexVLA to be end-to-end trained, as discussed in Appendix C.

#### 2.2.1 ALGORITHM DESIGN

We aim to enhance function-calling accuracy while simultaneously learning failure-recovery strategies. To this end, we develop a recovery-aware reward within the PPO (Schulman et al., 2017) algorithm. We first formalize the problem, then present the reward design, and finally illustrate the Cortex-PPO algorithm.

**Problem Formulation** We consider episodic RL for multi-step manipulation tasks specified by a task description $x \in \mathcal{X}$. At each step $t = 1, \cdots, T$, the agent observes $s_t \in \mathcal{S}$, outputs a structured function call $a_t \in \mathcal{A}$, receives reward $r_t \in \mathbb{R}$, and transitions to $s_{t+1}$. A trajectory is $\tau = \{(s_t, a_t, r_t)\}_{t=1}^{T}$. Since each sub-task involves multiple objects, we decompose $\tau$ into object-specific sub-trajectories: $\tau_i = \{(s_{i,t}, a_{i,t}, r_{i,t})\}_{t=1}^{|o_i|}$, where $|o_i|$ is the number of steps for object $o_i$. The agent follows a stochastic policy $\pi_\theta(a|s, x)$ with reward discounts $\gamma \in (0, 1]$.

**Reward Design** To stabilize fine-tuning for ultra-long-horizon tasks, we design a bounded, smooth, recovery-aware reward, *Cortex-Reward*. Let action correctness be $r_{i,t} \in \{-1, +1\}$, and define the failure recovery indicator

$$\psi_{i,t} = \begin{cases} 1, & \text{if } r_{i,t} = +1 \text{ and a valid recovery is executed,} \\ 0, & \text{otherwise.} \end{cases} \quad (1)$$

Set $z_{i,t} = r_{i,t} + \alpha\psi_{i,t}$, with $\alpha > 0$ controls the relative weighting of recovery. We then apply the hyperbolic tangent $\tanh$ to bound and smoothly scale $z_{i,t}$ to $(-1, 1)$, which stabilizes training and prevents large gradient spikes. To further mitigate reward sparsity and improve cross-environment generalization, we add independent Gaussian noise $\varepsilon_{i,t} \sim \mathcal{N}(0, \sigma^2)$, which is independent across time steps and independent of policy sampling. The final Cortex-Reward is

$$R_{i,t} = \tanh(\kappa z_{i,t}) + \varepsilon_{i,t}, \quad (2)$$

where $\kappa > 0$ scales the hyperbolic tangent contraction. For object $o_i$, the reward sequence is $R_i = (R_{i,1}, \ldots, R_{i,|o_i|})$.

**Cortex-PPO** We optimize with PPO (Schulman et al., 2017) using *Cortex-Reward* as the signal:

$$\mathcal{L}_{\text{Cortex-PPO}}(\theta) = \mathbb{E}_{\tau_i \sim \pi_{\theta_{\text{old}}}} \left[ \sum_{t=1}^{|o_i|} \min\left( \rho_{i,t}(\theta)\hat{A}_{i,t}, \text{clip}(\rho_{i,t}(\theta), 1 - \epsilon, 1 + \epsilon)\hat{A}_{i,t} \right) \right], \quad (3)$$

where $\rho_{i,t}(\theta) = \frac{\pi_\theta(a_{i,t}|s_{i,t})}{\pi_{\theta_{\text{old}}}(a_{i,t}|s_{i,t})}$. Using Generalized Advantage Estimation (GAE) (Schulman et al., 2015),

$$\hat{A}_{i,t} = \sum_{l=0}^{|o_i|-t} (\gamma\lambda)^l \delta_{i,t+l}, \qquad \delta_{i,t} = R_{i,t} + \gamma V_\phi(s_{i,t+1}) - V_\phi(s_{i,t}), \tag{4}$$

with $\lambda \in [0,1]$ and value function $V_\phi$. Cortex-PPO integrates recovery-aware rewards with noise injection and serves as a fine-tuning algorithm specifically designed for CortexVLA. We next provide theoretical analyses of this algorithm.

### 2.2.2 THEORETICAL ANALYSES

We focus on analyzing the unbiasedness and the generalization ability of Cortex-PPO. Full proofs of the theorems are provided in Appendix B.

**Assumption 1 (Deterministic critic).** The critic $V_\phi(s)$ is deterministic conditional on the observed state-action sequence $\bar{\tau}_i = \{(s_{i,t}, a_{i,t})\}_{t=1}^{|o_i|}$, i.e.,

$$\mathbb{P}\left(V_\phi(s) \in \cdot | \bar{\tau}_i, \{\varepsilon_{i,t}\}\right) = \mathbb{P}\left(V_\phi(s) \in \cdot | \bar{\tau}_i\right). \tag{5}$$

This assumption is standard in policy-gradient analysis, since critics are typically trained on observed states and actions only, without depending on additive reward noise.

**Theorem 1 (Unbiasedness under additive reward noise).** *Under the above assumption and the reward model in Equation 2, for any fixed $\bar{\tau}_i$:*

1. *(**Reward**) For every $t$,*

$$\mathbb{E}_\varepsilon[R_{i,t} \mid \bar{\tau}_i] = \tanh(\kappa z_{i,t}). \tag{6}$$

2. *(**GAE**) Let $\widehat{A}_{i,t}$ denote the advantage estimate computed from noisy rewards $R_{i,t}$, and $\widehat{A}_{i,t}^0$ the estimate with noiseless rewards $\tanh(\kappa z_{i,t})$. Then*

$$\mathbb{E}_\varepsilon[\widehat{A}_{i,t} \mid \bar{\tau}_i] = \widehat{A}_{i,t}^0. \tag{7}$$

3. *(**Policy Gradient**) Set $\widehat{g}(\theta; \bar{\tau}) = \sum_i \sum_t^{|o_i|} \nabla_\theta \log \pi_\theta(a_{i,t} \mid s_{i,t}) \, \widehat{A}_{i,t}.$ as the empirical GAE-based policy-gradient estimator. Then, by taking the expectation over the additive noises, we have*

$$\mathbb{E}_\varepsilon\left[\widehat{g}(\theta; \bar{\tau}) \mid \bar{\tau}\right] = \sum_i \sum_t^{|o_i|} \nabla_\theta \log \pi_\theta(a_{i,t} \mid s_{i,t}) \, \widehat{A}_{i,t}^0 = \widehat{g}^0(\theta; \bar{\tau}), \tag{8}$$

*where $\widehat{g}^0(\theta; \bar{\tau})$ denotes the policy-gradient estimator computed with noiseless rewards.*

Therefore, the Cortex-Reward in Equation 2 is unbiased. For any object-specific trajectory, the GAE-based policy-gradient estimator computed with noisy rewards has the same expectation as with noiseless rewards. More broadly, this unbiasedness holds for any additive zero-mean noise, regardless of its distribution. As a result, noise injection preserves the validity of the learning signal and ensures that Cortex-PPO optimizes the same objective as in the noiseless case. We now analyze its generalization behavior and establish an information-theoretic bound that characterizes cross-environment performance guarantees.

Let $\mathcal{E}$ denote the distribution over environments, and $E \sim \mu$ be a random environment drawn from $\mathcal{E}$. For each $e \in \mathcal{E}$, let $J_e$ denote the expected return in environment $e$ under policy $\pi_\theta$, with the aggregated reward along a trajectory approximately bounded in $[a, b]$. Let $R$ be the observed reward under the noisy Cortex-Reward model.

**Theorem 2 (Information-Theoretic Bound on Cross-Environment Performance).** *Let $E$ be a random environment drawn from $\mu$, and $R$ be the observed reward under the external-noise Cortex-Reward model. Then the expected absolute performance difference between two independent environment samples $E, E'$ satisfies*

$$\mathbb{E}_{E,E'}\left[|J_E - J_{E'}|\right] \leq (b-a)\sqrt{2\,I(E;R)}, \tag{9}$$

*where $I(E; R)$ denotes the mutual information between the environment index $E$ and the observed rewards $R$.*

This theorem shows that cross-environment variation in expected returns is controlled by the mutual information $I(E; R)$ between the environment identity and observed rewards. Under the noisy Cortex-Reward $R = f(z_\tau) + \varepsilon$ with $\varepsilon \perp (E, \tau)$, the data-processing inequality (Beaudry & Renner, 2012) implies $I(E; R) = I(E; f(z_\tau) + \varepsilon) \leq I(E; f(z_\tau))$. Consequently, as the noise variance grows, $I(E; R)$ is non-increasing and typically strictly decreasing. By Theorem 2, this reduction in $I(E; R)$ tightens the cross-environment performance bound. This demonstrates that noise injection weakens environment-specific information in rewards, thereby enhancing cross-environment generalization capabilities and providing a principled mechanism for achieving more robust policy performance across diverse environments.

## 3 EXPERIMENTS

### 3.1 MAIN EXPERIMENT – ULTRA-LONG-HORIZON TASK

The main experiment we designed for examining the effectiveness of CortexVLA and several baseline models can be referred to as *ultra-long-horizon* task. We first provide the task definitions and metrics of this experiment. Then, we introduce the baselines we selected for evaluation. Finally, we present the experiment results and analyses.

#### 3.1.1 TASK DEFINITIONS AND METRICS

The *ultra-long-horizon* task requires models to sequentially locate, grasp, and release multiple target objects in the exact order specified by user instructions. We use the term *ultra-long* because these tasks can, in principle, extend indefinitely, constrained only by hardware, environment, and time. In our experiment, we set the maximum length to 14 sub-tasks, which already poses a severe challenge for existing methods. Note that a *sub-task* here refers to a complete small task, such as locating, grasping, and releasing an object, rather than a decomposed action. Success in this experimental setting requires accurate action execution over extended durations, as well as precise decision-making and consistent task memory. These capabilities are difficult for current methods to achieve.

Concretely, each task starts with a natural-language instruction specifying the required order of target objects, and the model must follow this sequence precisely. Details of the instruction design are provided in Appendix E. Given the complexity of these tasks, we evaluate performance with two metrics: *success rate* for overall performance and *average success length* for fine-grained capability.

#### 3.1.2 BASELINES

We evaluate CortexVLA against representative baseline models, grouped into two categories: the end-to-end methods and the hierarchical methods.

**End-to-end methods** Octo (Octo Model Team et al., 2024) is a lightweight transformer-based model that accepts language commands and goal images. **OpenVLA** (Kim et al., 2024) builds on a 7B Llama2 (Touvron et al., 2023), integrating DinoV2 (Oquab et al., 2023) and SigLip (Zhai et al., 2023) for multimodal understanding, with actions expressed as discrete tokens. **ACT** (Zhao et al., 2023) targets bimanual manipulation using a Transformer and VAE with Action Chunk and Temporal Ensemble for precise control. $\boldsymbol{\pi_0}$ (Black et al., 2024), trained on large-scale teleoperation data with PaliGemma (Beyer et al., 2024), can handle complex tasks such as cloth folding.

**Hierarchical methods** RoBridge (Zhang et al., 2025) adopts a three-layer architecture with a high-level cognitive planner (HCP), an invariant operable representation (IOR), and a guided embodied agent (GEA). The HCP decomposes instructions into primitive actions and generates IOR, which encodes depth, masks, action types, and constraints. IOR is updated at different frequencies and serves as input to the GEA, which executes the actions to complete the task. All three layers are realized by GPT-4o (Hurst et al., 2024). **VLA-OS** (Gao et al., 2025) provides modular architectures for action-only (A), integrated (I), and hierarchical (H) paradigms. Specifically for the VLA-OS-H, it uses the VLM together with planning heads for task planning, and modifies the action head to an encoder-decoder transformer for policy learning. Note that the action execution parts of these two methods are either closed-source or difficult to reproduce, so we only evaluate their planning

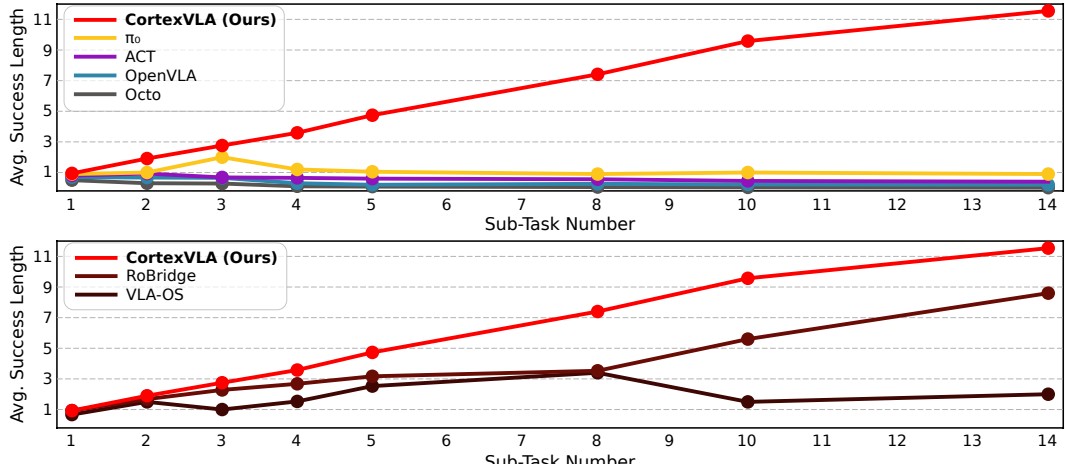

Figure 2: **Ultra-long-horizon experiment results (average success length).** We compare the CortexVLA with baselines. The upper panel reports results against end-to-end methods, while the lower panel reports results against hierarchical methods.

Table 1: **Ultra-long-horizon experiment results (success rate).** The mark "failed" indicates that no successful trials were observed. Both Octo and OpenVLA achieved no successes in this experiment.

| Number of Sub-tasks | ACT | $\pi_0$ | VLA-OS | RoBridge | CortexVLA (w/o Cortex-PPO) | CortexVLA |
|---|---|---|---|---|---|---|
| 1 | 80.00% | 90.00% | 33.33% | 93.33% | 93.75% | **94.12%** |
| 2 | 13.33% | 66.67% | 33.33% | 84.21% | 87.50% | **90.91%** |
| 3 | 8.00% | 45.00% | failed | 72.22% | 85.71% | **88.24%** |
| 4 | failed | failed | failed | 52.63% | 85.29% | **86.49%** |
| 5 | failed | failed | failed | 43.47% | 84.85% | **85.71%** |
| 8 | failed | failed | failed | failed | 83.78% | **84.38%** |
| 10 | failed | failed | failed | failed | 77.42% | **80.65%** |
| 14 | failed | failed | failed | failed | 58.33% | **72.73%** |
| **Average** | 12.67% | 25.21% | 8.33% | 43.23% | 82.08% | **85.40%** |

abilities. This assumes perfect action experts for these methods, which gives them extra advantages. Note that the planning stage of these two methods is not limited to scheduling the order of sub-tasks; it also requires determining whether each sub-task has been successfully completed.

### 3.1.3 RESULTS AND ANALYSES

From Table 1, we can observe that CortexVLA achieves an average of 85.40% success rate across the entire experiments, which is more than 3 times higher than $\pi_0$ and nearly 2 times higher than planning-only RoBridge. This shows the strong stability of CortexVLA. From Figure 2, we can observe that as the task instruction length increases, the average success length of CortexVLA grows almost linearly and consistently approaches the total task length. The average task length of CortexVLA in the hardest setting of 14 sub-tasks is 11.55, which shows a huge gap with baseline methods. We also calculated the linear regression coefficient $k$ for the CortexVLA results in Figure 2, obtaining $k = 0.8519$, which is closer to the ideal value of 1. This indicates that CortexVLA scales stably with task length and suffers little performance degradation when facing error accumulation.

For the end-to-end baselines, $\pi_0$ achieves a relatively higher average success rate of 25.21%, but it fails to perform any successful task once the number of sub-tasks reaches 4 or more. In terms of average success length, these methods show linear growth only within short horizons, after which performance plateaus or even declines. For hierarchical methods that only evaluate planning ability, there still remains a substantial performance gap compared to the full CortexVLA system.

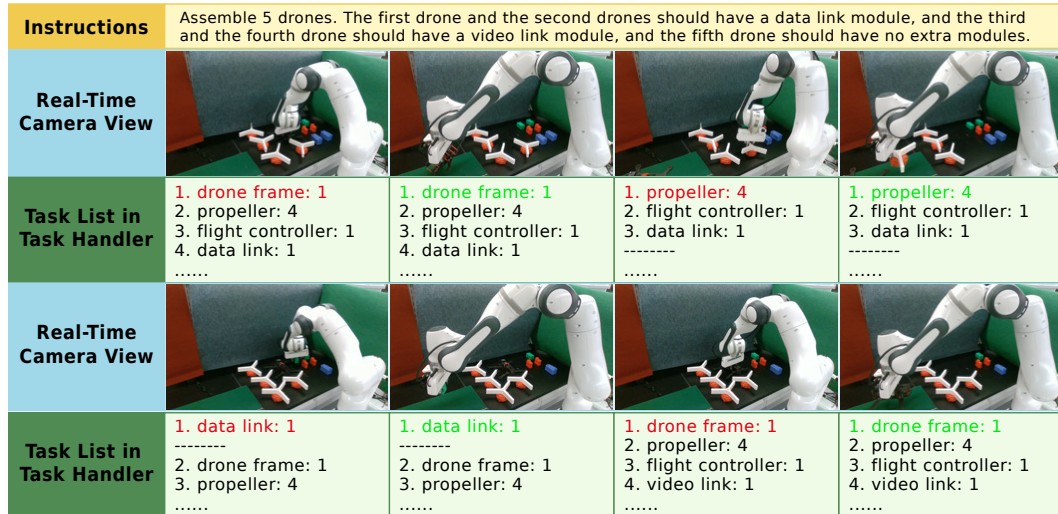

Figure 3: **Examples of CortexVLA performing flexible manufacturing tasks.** The figure shows real-time camera views during task execution, with the corresponding task lists maintained by the *task handler*. Tasks assigned to different drones are separated by dashes. CortexVLA always operates on the first sub-task in the list: red highlights the sub-task currently being executed, and green highlights a completed sub-task that will soon be removed or decremented. For clarity, we present representative examples rather than the full execution sequence.

Table 2: **Experiment results of the flexible manufacturing scenario.**

| Number of Drones | Avg. Number of Sub-tasks | Avg. Succ. Len. | Succ. Rate |
|---|---|---|---|
| 1 drone | 6 | 6.00 | 100.00% |
| 2 drones | 13 | 12.94 | 94.12% |
| 3 drones | 19 | 17.30 | 86.96% |
| 5 drones | 31 | 26.69 | 81.25% |
| **Avg. Succ. Rate** | | | **90.58%** |

The primary reason for the strong performance of CortexVLA lies in its modular coordination mechanism and task state memory capability. The former helps prevent error propagation between perception and action, while the latter ensures that each sub-task can be executed independently. Additionally, Cortex-PPO equips CortexVLA with error recovery capabilities, further enhancing its performance (see Table 1). In comparison, end-to-end baselines face challenges in mitigating error accumulation. And since our experiments included repetitive subtasks, this confused training for end-to-end methods, further exacerbating performance degradation. While hierarchical baseline methods only evaluate planning performance, they still exhibit errors during long-horizon task planning and often fail in progress tracking during execution. Specifically, RoBridge frequently misjudges task progress, particularly incorrectly detecting whether the gripper has successfully grasped an object. VLA-OS-H, in turn, often plans an unnecessary additional object or assigns incorrect object names to sub-tasks. Furthermore, due to the limited stability and repeatability of baseline methods, their performance on complex tasks cannot be guaranteed. In contrast, our approach demonstrates robust stability and straightforward reproducibility.

## 3.2 CASE STUDIES

To further evaluate the generalization ability and adaptation ability of CortexVLA, we conduct case studies across different scenarios. Below, we present two representative adaptations, with additional experiments provided in Appendix F.

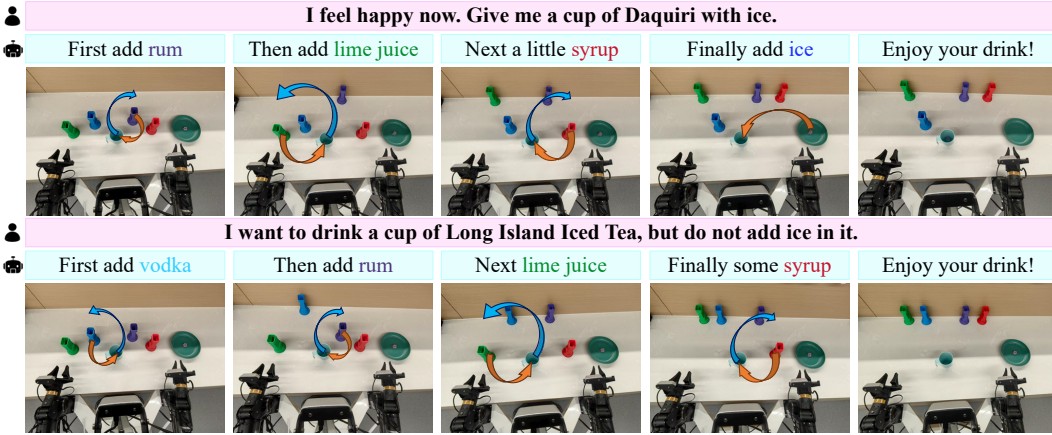

Figure 4: **Examples of CortexVLA serving customers as a bartender.** Customer requests are shown in pink, while the decomposed sub-tasks are shown in cyan with corresponding operation diagrams below. The orange arrows in the diagrams mean adding ingredients, and the blue arrows mean putting the containers back.

### 3.2.1 SCENARIO ADAPTATION – FLEXIBLE MANUFACTURING

Flexible manufacturing is a production method that can quickly adapt to changes in product type or demand. Inspired by this, we design a drone componentsorting task that mimics a common assembly-line scenario. The VLA must control a robot to sequentially place drone components onto a conveyor belt according to specified assembly requirements. A typical drone requires a drone frame, four motors with propellers, and a flight controller. Depending on the configuration, some drones also require a data link module, a video link module, or both. Even assembling only five drones involves more than 30 sub-tasks, making this task a substantial challenge for existing VLA models. To evaluate the generalization ability of CortexVLA, we directly adapt the model from the main experiment to this scenario, modifying only the prompt without any additional fine-tuning.

We assess performance on instructions for assembling one to five drones, including tasks that combine different drone configurations (see Appendix E). Results are shown in Table 2. CortexVLA achieves a 90.58% average success rate across all experiments and sustains an 81.25% success rate with an average success length of 26.69 on the most difficult setting with 31 sub-tasks. These results highlight CortexVLAs stability and strong generalization ability for ultra-long-horizon tasks across diverse scenarios. Representative examples are shown in Figure 3.

### 3.2.2 SCENARIO ADAPTATION – BARTENDER

In this scenario, we adapt CortexVLA to function as a bartender, preparing cocktails based on customer requests. To approximate real-world conditions, requests are given as variable natural language commands that typically specify only cocktail names and personal preferences (e.g., whether to add ice or syrup). This setting is particularly challenging because it requires recalling cocktail recipes while also accommodating individual preferences. Note that cocktail recipes typically have strict requirements for the order in which ingredients are added.

Figure 4 illustrates examples of CortexVLA making cocktails. The *planner* of the Central Cortex decomposes diverse natural language instructions into sequential sub-tasks according to the recipes and the customers' preferences. By employing ACT (Zhao et al., 2023) as both the Visual Cortex and the Motor Cortex, CortexVLA achieves an average success rate of 91.67% in this case study. These results highlight both the adaptability of CortexVLA to new scenarios and the flexibility of replacing tool libraries and modules.

## 4 CONCLUSION

In this paper, we introduce CortexVLA, a novel VLA paradigm that bridges the gap between cognition and action through LLM-based function calling. The framework consists of a Central Cortex for planning and function orchestration, a Visual Cortex for perception, and a Motor Cortex for control. We further propose Cortex-PPO, a recovery-aware RL algorithm that enhances the capabilities of CortexVLA. Our theoretical analyses establish the soundness and generalization guarantees of this algorithm. Extensive experiments demonstrate that CortexVLA not only surpasses strong baselines but also adapts robustly across diverse application scenarios. In the future, we will further advance the capabilities of the three Cortices and explore broader domains to extend the versatility of CortexVLA.

## REPRODUCIBILITY STATEMENT

To ensure reproducibility, we provide implementation details, including model selection, prompt design, and fine-tuning details in Appendix D and user instructions in Appendix E. Full training and inference codes will be released upon the paper acceptance.

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

## A  RELATED WORK

**Vision-Language-Action Models**  Recent explorations in the field of robotic manipulation have made significant progress (Zhao et al., 2023; Brohan et al., 2022; Fu et al., 2024), with the VLA model emerging as a promising direction. (Black et al., 2024; Kim et al., 2024; Octo Model Team et al., 2024; Zitkovich et al., 2023; Wen et al., 2025b;a; Zhou et al., 2025; Zhao et al., 2025a; Zhang et al., 2024; Zhen et al., 2024; Belkhale et al., 2024; Zheng et al., 2024; Zhao et al., 2025b; Chen et al., 2025; Fan et al., 2025; Chen et al., 2025; Li et al., 2025; Wang et al., 2025; Physical Intelligence et al., 2025; Zhai et al., 2025). Most approaches leverage pretrained VLMs to process multimodal information, further training them on large-scale robotic datasets (O'Neill et al., 2024; Bu et al., 2025; Walke et al., 2023; Lee et al., 2020; Zhu et al., 2022) to generate action outputs. However, these models generally follow a Markov decision-making paradigm, which limits their ability to solve complex long-horizon tasks. They often encounter decision bottlenecks when repetitive or similar actions are required, and collecting continuous operational data for long-horizon tasks is both time-consuming and prone to errors, with failures accumulating over time. Although hierarchical VLA architectures (Zhang et al., 2025; Shi et al., 2025; Gao et al., 2025) have shown improvements in long-horizon task execution, they still suffer from imprecise planning, limited task-state tracking, or heavy reliance on synthetic training data. In contrast, our approach introduces robust planning and memory mechanisms that sustain high success rates even in extremely long-horizon tasks. Furthermore, it supports seamless integration with both traditional algorithms and VLA models, featuring hot-swappable modularity that allows new skills to be added or replaced with minimal overhead.

**Function Calling Studies**  Integrating external tools can expand the capability boundaries of LLMs, enabling them to address specialized and high-precision tasks (Qin et al., 2023; Liu et al., 2024b). Remarkably, with only tool descriptions and usage examples provided in prompts, LLMs can already invoke tools without fine-tuning (Yao et al., 2023; Ruan et al., 2023; Hsieh et al., 2023). Among these, the well-known ReAct (Yao et al., 2023) method enables LLMs to solve complex tasks through alternating cycles of reasoning and action. However, due to the high dependence of such methods on the model's initial capabilities, their effectiveness and scope of application are limited. Further explorations achieve remarkable improvements through fine-tuning on tool-oriented datasets, which allows LLMs to perform reliably even in scenarios requiring complex planning or the use of unfamiliar tools (Liu et al., 2024b; Schick et al., 2023; Tang et al., 2023; Patil et al., 2024; Qin et al., 2023; Abdelaziz et al., 2024; Liu et al., 2024c). Among these, ToolACE (Liu et al., 2024b) implements an automated data pipeline for function calls to overcome the limitations of relying on existing APIs. Building on these advances, our approach fully exploits the function-calling capabilities of LLMs to bridge vision and action, thereby enabling more flexible and robust robotic intelligence.

## B  DETAILS OF CORTEX-PPO AND THEORETICAL ANALYSES

### B.1  UNBIASEDNESS UNDER ADDITIVE REWARD NOISE

We restate Theorem 1 and provide its proof here.

**Theorem 3 (Restatement of Theorem 1).** *Under the Assumption 1 and the reward model in Equation 2, for any fixed $\bar{\tau}_i$:*

1. *(**Reward**) For every $t$,*

$$\mathbb{E}_\varepsilon[R_{i,t} \mid \bar{\tau}_i] = \tanh(\kappa z_{i,t}). \tag{10}$$

2. *(**GAE**) Let $\widehat{A}_{i,t}$ denote the advantage estimate computed from noisy rewards $R_{i,t}$, and $\widehat{A}_{i,t}^0$ the estimate with noiseless rewards $\tanh(\kappa z_{i,t})$. Then*

$$\mathbb{E}_\varepsilon[\widehat{A}_{i,t} \mid \bar{\tau}_i] = \widehat{A}_{i,t}^0. \tag{11}$$

3. *(**Policy Gradient**) Set $\widehat{g}(\theta; \bar{\tau}) = \sum_i \sum_t^{|o_i|} \nabla_\theta \log \pi_\theta(a_{i,t} \mid s_{i,t}) \, \widehat{A}_{i,t}$. as the empirical GAE-based policy-gradient estimator. Then, by taking the expectation over the additive noises, we*

*have*

$$\mathbb{E}_\varepsilon\big[\,\widehat{g}(\theta;\bar{\tau})\mid\bar{\tau}\,\big] \;=\; \sum_i\sum_t^{|o_i|}\nabla_\theta\log\pi_\theta(a_{i,t}\mid s_{i,t})\,\widehat{A}^0_{i,t} \;=\; \widehat{g}^0(\theta;\bar{\tau}),\tag{12}$$

*where $\widehat{g}^0(\theta;\bar{\tau})$ denotes the policy-gradient estimator computed with noiseless rewards.*

*Proof.*  1. **(Reward)** Taking conditional expectation and using linearity of expectation together with $\mathbb{E}[\varepsilon_{i,t}]=0$, we obtain

$$\mathbb{E}_\varepsilon[R_{i,t}\mid\bar{\tau}_i] = \mathbb{E}_\varepsilon[\tanh(\kappa z_{i,t})+\varepsilon_{i,t}\mid\bar{\tau}_i] = \tanh(\kappa z_{i,t})+\mathbb{E}_\varepsilon[\varepsilon_{i,t}\mid\bar{\tau}_i] = \tanh(\kappa z_{i,t}).\tag{13}$$

2. **(GAE)** Consider the GAE computed from cortex-reward, its conditional expectation given the stateaction sequence $\bar{\tau}_i$ is

$$\begin{aligned}
\mathbb{E}_\varepsilon[\widehat{A}_{i,t}\mid\bar{\tau}_i] &= \mathbb{E}_\varepsilon\Big[\sum_{\ell=0}^{|o_i|-t}(\gamma\lambda)^\ell\,\delta_{i,t+\ell}\;\Big|\;\bar{\tau}_i\Big]\\
&= \sum_{\ell=0}^{|o_i|-t}(\gamma\lambda)^\ell\,\mathbb{E}_\varepsilon[\delta_{i,t+\ell}\mid\bar{\tau}_i]\\
&= \sum_{\ell=0}^{|o_i|-t}(\gamma\lambda)^\ell\,\mathbb{E}_\varepsilon[R_{i,t+\ell}+\gamma V_\phi(s_{i,t+\ell+1})-V_\phi(s_{i,t+\ell})\mid\bar{\tau}_i]\\
&= \sum_{\ell=0}^{|o_i|-t}(\gamma\lambda)^\ell\Big(\mathbb{E}_\varepsilon[R_{i,t+\ell}\mid\bar{\tau}_i]+\gamma V_\phi(s_{i,t+\ell+1})-V_\phi(s_{i,t+\ell})\Big)\\
&= \sum_{\ell=0}^{|o_i|-t}(\gamma\lambda)^\ell\Big(\tanh(\kappa z_{i,t+\ell})+\gamma V_\phi(s_{i,t+\ell+1})-V_\phi(s_{i,t+\ell})\Big)\\
&= \sum_{\ell=0}^{|o_i|-t}(\gamma\lambda)^\ell\,\delta^0_{i,t+\ell}\\
&= \widehat{A}^0_{i,t}.\tag{14}
\end{aligned}$$

3. **(Policy gradient)** Conditional on $\bar{\tau}$, the score function $\nabla_\theta\log\pi_\theta(a_{i,t}\mid s_{i,t})$ is deterministic. Therefore,

$$\begin{aligned}
\mathbb{E}_\varepsilon\big[\widehat{g}(\theta;\bar{\tau})\mid\bar{\tau}\big] &= \mathbb{E}_\varepsilon\Big[\sum_i\sum_t\nabla_\theta\log\pi_\theta(a_{i,t}\mid s_{i,t})\,\widehat{A}_{i,t}\;\Big|\;\bar{\tau}\Big]\\
&= \sum_i\sum_t\nabla_\theta\log\pi_\theta(a_{i,t}\mid s_{i,t})\,\mathbb{E}_\varepsilon[\widehat{A}_{i,t}\mid\bar{\tau}]\\
&= \sum_i\sum_t\nabla_\theta\log\pi_\theta(a_{i,t}\mid s_{i,t})\,\mathbb{E}_\varepsilon[\widehat{A}_{i,t}\mid\bar{\tau}_i]\\
&= \sum_i\sum_t\nabla_\theta\log\pi_\theta(a_{i,t}\mid s_{i,t})\,\widehat{A}^0_{i,t}\\
&= \widehat{g}^0(\theta;\bar{\tau}).\tag{15}
\end{aligned}$$

□

## B.2  INFORMATION-THEORETIC GENERALIZATION ANALYSIS OF EXTERNAL CORTEX NOISE

We first expand all notation and definitions, and then provide a complete proof of Theorem 2.

Let $\mathcal{E}$ denote the distribution over environments, and for each $e\in\mathcal{E}$, let $\tau$ be a trajectory sampled according to policy $\pi_\theta$ with distribution $p_e(\tau)$. Consider the external-noise cortex-reward defined by

$$R\mid\tau\sim\mathcal{N}(\tanh(\kappa z_\tau),\sigma^2),\tag{16}$$

where $z_\tau$ denotes the task-specific signal aggregated along $\tau$, $\kappa > 0$ is a scaling parameter, and $\sigma^2$ is the variance of the additive Gaussian noise. The marginal reward distribution in environment $e$ is

$$p_e(R) = \int p_e(\tau)\, p(R \mid \tau)\, d\tau, \tag{17}$$

and the overall marginal across environments is

$$p(R) = \int \mu(e)\, p_e(R)\, de, \tag{18}$$

where $\mu(e)$ represents the environment distribution.

The expected return in environment $e$ is defined as

$$J_e \triangleq \mathbb{E}_{\tau \sim p_e, R \sim p(R|\tau)}[R(\tau)], \tag{19}$$

where $R(\tau)$ denotes the aggregated reward along $\tau$ and is approximately bounded, i.e., with high probability $R(\tau) \in [a, b]$ for small $\sigma^2 > 0$.

**Theorem 4** (**Restatement of Theorem 2**). *Let $E$ be a random environment drawn from $\mu$, and $R$ be the observed reward under the external-noise Cortex-Reward model. Then the expected absolute performance difference between two independent environment samples $E, E'$ satisfies*

$$\mathbb{E}_{E,E'}\big[\, |J_E - J_{E'}| \,\big] \le (b - a)\sqrt{2\, I(E; R)}, \tag{20}$$

*where $I(E; R)$ denotes the mutual information between the environment index $E$ and the observed rewards $R$.*

*Proof.* For any pair of environments $e, e'$, we have

$$|J_e - J_{e'}| = \left| \int R(\tau)\, (p_e(R) - p_{e'}(R))\, dR \right|$$

$$\le \int \left| R(\tau) - \frac{a+b}{2} \right| |p_e(R) - p_{e'}(R)|\, dR$$

$$\le (b - a)\, \mathrm{TV}(p_e(R), p_{e'}(R)), \tag{21}$$

where $\mathrm{TV}(p, q) = \frac{1}{2} \int |p(R) - q(R)|\, dR$ denotes the total variation distance. Introducing the marginal reward distribution across environments. Applying the triangle inequality for total variation gives

$$\mathrm{TV}(p_e(R), p_{e'}(R)) \le \mathrm{TV}(p_e(R), p(R)) + \mathrm{TV}(p_{e'}(R), p(R)). \tag{22}$$

By Pinsker's inequality (Cover & Thomas, 2006), $\mathrm{TV}(p, q) \le \sqrt{\frac{1}{2} D_{\mathrm{KL}}(p\|q)}$, which implies

$$|J_e - J_{e'}| \le (b - a)\left( \sqrt{\frac{1}{2} D_{\mathrm{KL}}(p_e(R)\|p(R))} + \sqrt{\frac{1}{2} D_{\mathrm{KL}}(p_{e'}(R)\|p(R))} \right). \tag{23}$$

Taking the expectation over independent environments $E, E' \sim \mu$ and using the linearity of expectation, we obtain

$$\mathbb{E}_{E,E'}[|J_E - J_{E'}|] \le 2(b - a)\, \mathbb{E}_E\left[ \sqrt{\frac{1}{2} D_{\mathrm{KL}}(p_E(R)\|p(R))} \right]. \tag{24}$$

Applying Jensen's inequality to the concave square-root function yields

$$\mathbb{E}_E\left[ \sqrt{\frac{1}{2} D_{\mathrm{KL}}(p_E(R)\|p(R))} \right] \le \sqrt{\frac{1}{2} \mathbb{E}_E[D_{\mathrm{KL}}(p_E(R)\|p(R))]}. \tag{25}$$

Finally, by the definition of mutual information,

$$I(E; R) = \mathbb{E}_E[D_{\mathrm{KL}}(p_E(R)\|p(R))], \tag{26}$$

we conclude that

$$\mathbb{E}_{E,E'}[|J_E - J_{E'}|] \le (b - a)\sqrt{2\, I(E; R)}, \tag{27}$$

which completes the proof. $\square$

## C  END-TO-END RL FOR TRAINING CORTEXVLA

The modular design of CortexVLA can largely improve the generalizability and interpretability over previous VLA models. However, this may restrict the flexibility of CortexVLA if it cannot be end-to-end trained. To overcome this limitation, we propose an end-to-end RL method, improved from the policy gradient method. The gradient of the proposed method can be written as follows:

$$
\begin{aligned}
g &= \mathbb{E}[\sum_{t=0}^{T} A_t \nabla_\theta \log \pi_\theta(a_t|s_t)] \\
&= \mathbb{E}[\sum_{t=0}^{T} A_t \nabla_\theta \log \pi_{\theta_{\text{action}}}(a_t|s_t, o_{\text{central}}) \pi_{\theta_{\text{central}}}(o_{\text{central}}|s_t)] \\
&= \mathbb{E}[\sum_{t=0}^{T} A_t \nabla_\theta (\log \pi_{\theta_{\text{action}}}(a_t|s_t, o_{\text{central}}) + \log \pi_{\theta_{\text{central}}}(o_{\text{central}}|s_t))]
\end{aligned}
\tag{28}
$$

where $A_t$ is the advantage, $\theta_{\text{action}}$ is the Motor Cortex's model's parameter, $a_t$ is the model's action output, $o_{\text{central}}$ is the Central Cortex's output, $\theta_{\text{central}}$ is the Central Cortex's parameter. Here, we do not include the parameter of Visual Cortex into this formulation. This does not mean that the end-to-end RL cannot optimize Visual Cortex. In our implementation, we generally employed production-ready modules, e.g., VLA models like $\pi_0$, as both Visual Cortex and Motor Cortex. As a standard operation in existing VLA works, we fine-tune only the action expert components, which are related to the Motor Cortex. Unfreezing the VLM (Visual-Cortex-related) for end-to-end RL often destabilizes training and is not the typical VLA-finetuning protocol. By this formulation, we are able to train the CortexVLA end-to-end. Besides, by this formulation, we can also incorporate other existing real-world robotics reinforcement learning implementations for training the Motor Cortex (Luo et al., 2024). This can further enable our framework for dexterous manipulations and demonstrates the extendability of the proposed framework.

We evaluated this end-to-end algorithm on dexterous manipulation. With $\pi_0$ as the Motor Cortex, CortexVLA achieved a 53.85% success rate without end-to-end training. After end-to-end training, the success rate increased to 69.23%, indicating a substantial improvement. These results demonstrate that CortexVLA can be trained end-to-end.

## D  IMPLEMENTATION DETAILS

In this section, we present the implementation details of the CortexVLA, including the model selection, prompt design, and SFT details.

### D.1  MODEL SELECTION

As mentioned in Section 2.1, the Central Cortex contains two LLM-based decision layers, the *Planner* and the *Allocator*. For the *Planner*, its primary role is to understand user instructions and generate a task list. There are numerous model options available. Based on our current exploration, the smallest viable model is Qwen3-0.6B (requiring some fine-tuning). However, due to its limited parameters, it struggles to comprehend more complex instructions. For handling more sophisticated instructions, larger models such as GPT-4o can be used to generate task lists more reliably. The *Allocator* is implemented with Qwen3-8B, which offers strong natural language understanding and tool invocation capabilities while maintaining a parameter scale that is still practical for fine-tuning.

### D.2  PROMPTING TEMPLATES

We present the example of system prompts for guiding the *Planner* and the *Allocator* in Figure 5 and Figure 6, respectively.

```
You are a helpful assistant in a drone manufacturing factory.
To assemble a drone, one drone frame and four propellers is always necessary, and the optional modules
are a flight controller, a data link and a video link.
You need to make a task list according to user's assembling requirements.
To make a task list, you need to add all modules a drone needs as a subtask, and then add the modules
for the next drone as the next task. Add drone id before the task list for each drone.
Note, all the tasks are in order, never merge two operation numbers when there is any other object
between them. Do not add blank line or extra lines in the task list.
Always output the full task list at once.
For example:
User Instruction: Assemble a drone with a flight controller and a data link
Your output:
1. drone frame: 1
2. propeller: 4
3. flight controller: 1
4. data link: 1

User Instruction: Assemble a drone with a video link, and another drone without any extra modules, and
a third one with a data link and a video link
Your output:
1. drone frame: 1
2. propeller: 4
3. video link: 1
--------
4. drone frame: 1
5. propeller: 4
--------
6. drone frame: 1
7. propeller: 4
8. data link: 1
9. video link: 1
```

Figure 5: Example of the system prompt for the *Planner*.

```
You are a helpful assistant.
Your have to use the provided tools to complete the given tasks.

# General Rules
1. You must only call one function per step. Do not combine multiple function calls in the same step.
2. If a function execution fails, follow its restart policy exactly as specified in its description.
3. If you get a prompt like "All Done" or "Nothing remain to do" that indicates the task is finished,
you must stop immediately and reply with: 'All Done'.
4. You should always follow the task list given after "Task remain to do:". The list is strictly ordered.
You must execute tasks in order without skipping or reordering. If multiple tasks remain, only focus on
the first task at each step.

# Tools

You may call one or more functions to assist with the user query.

You are provided with function signatures within <tools></tools> XML tags:
<tools>
{"type": "function", "function": {"name": <func_1>, "description": <desc_1>, "parameters": <param_1>}
{"type": "function", "function": {"name": <func_2>, "description": <desc_2>, "parameters": <param_2>}
{"type": "function", "function": {"name": <func_3>, "description": <desc_3>, "parameters": <param_3>}
{"type": "function", "function": {"name": <func_4>, "description": <desc_4>, "parameters": <param_4>}
</tools>

For each function call, return a json object with function name and arguments within
<tool_call></tool_call> XML tags:
<tool_call>
{"name": <function-name>, "arguments": <args-json-object>}
</tool_call>
```

Figure 6: Example of the system prompt for the *Allocator*.

## D.3 SFT DETAILS

We fine-tuned the Allocator in CortexVLA using LoRA on our self-generated function-call training data. The LoRA configuration employed a LoRA rank of 16, $\alpha = 32$, and dropout set as 0.1. Training was performed with a batch size of 4, a learning rate of $5 \times 10^{-5}$, and one epoch is enough. A linear learning-rate scheduler with 10% warmup steps was applied.

# E  INSTRUCTIONS OF THE EXPERIMENTS

## E.1  INSTRUCTIONS OF THE MAIN EXPERIMENTS

The instructions of the main experiments (Section 3.1) are listed below:

- Instructions with 1 sub-task
  - Grab the orange toy and then release it.
  - Clasp the yellow tape measure momentarily before releasing.
  - Grasp the green cup and then release it.

- Instructions with 2 sub-tasks
  - First grab the orange toy and release it, then grab the yellow tape measure and release it.
  - Grasp and release the orange toy. Do the above operations twice.
  - Pick up the yellow tape measure and let go, then pick up the green cup and let go.

- Instructions with 3 sub-tasks
  - First grab the orange toy and release it, then grab the yellow tape measure and release it, finally grab the green cup and release it.
  - Operation sequence: First the green cup, then the green bowl, finally the orange toy. Perform grasp-release on each.
  - Grasp the orange toy and release it repeatedly for three times.

- Instructions with 4 sub-tasks
  - First grasp the orange toy and release it, then grasp the green cup and release it, next grasp the yellow tape measure and release it, finally grasp the orange toy again and release it
  - Operation sequence: First the yellow tape measure, then the orange toy twice, finally the green bowl. Grasp and release each.
  - Initiate capture and release cycles for the yellow tape measure, followed by the green cup, then the orange toy, and concluding with the white cube.

- Instructions with 5 sub-tasks
  - First grab the orange toy and release it. Then grab the yellow tape measure and release it. Next grab the orange toy again and release it. After that grab the green bowl and release it. Finally grab the green bowl again and release it.
  - Step 1: Grasp and release the yellow tape measure. Step 2: Grasp and release the toy. Step 3: Grasp and release the green cup. Step 4: Grasp and release the orange toy. Step 5: Grasp and release the yellow tape measure.
  - Initiate five sequential grasp-release cycles: Orange toy for three times and yellow tape measure twice.

- Instructions with 8 sub-tasks
  - Step 1: Grasp and release the orange toy. Step 2: Grasp and release the green bowl. Step 3: Grasp and release the yellow tape measure. Step 4: Grasp and release the white cube. Step 5: Grasp and release the green cup. Step 6: Grasp and release the green bowl. Step 7: Grasp and release the orange toy. Step 8: Grasp and release the yellow tape measure.
  - Begin with the orange toy: grasp then release. Follow with the yellow tape measure: grasp then release. Then the orange toy again: grasp then release. Next the green bowl: grasp then release. Continue to the white cube: grasp then release. Proceed to the green cup: grasp only. Then the orange toy once again: grasp then release. Conclude with the yellow tape measure: grasp then release.

- – Execute these grasp-release tasks sequentially: Task 1: Grasp the orange toy and then release, do operations above for three times. Task 2: The yellow tape measure for four times. End with grasping the white cube and then releasing it.

- Instructions with 10 sub-tasks

  - – Perform in order: 1) Grab the yellow tape measure and release it, 2) Grab the green cup and release it, 3) Grab the orange toy and release it, 4) Grab the orange toy and release it, 5) Grab the green bowl and release it, 6) Grab the yellow tape measure and release it, 7) Grab the orange toy and release it, 8) Grab the green cup and release it, 9) Grab the orange toy and release it, 10) Grab the green bowl and release it.

  - – Execute these grasp-release tasks sequentially: Task 1: Grasp the orange toy and then release, do operations above for 6 times. Task 2: The yellow tape measure for three times. End with grasping the white cube and then releasing it.

- Instructions with 14 sub-tasks

  - – Execute these grasp-release tasks sequentially: Task 1: Grasp the orange toy and then release, do operations above 3 times. Task 2: The yellow tape measure twice. Task 3: Grasp and then release the green cup. Task 4: The orange toy again for 4 times. Task 5: The green cup, then the yellow tape measure, then the white cube. Each operates once. End with grasping the orange toy and then releasing it.

### E.2 INSTRUCTIONS OF THE FLEXIBLE MANUFACTURING CASE STUDY

- Instructions with 1 drone

  - – Assemble a drone with a pilot controller

  - – Build one drone with a data link, only leave off video links and pilot controllers

  - – Assemble one drone equipped solely with a data link

- Instructions with 2 drones

  - – Assemble two drones; the first should be equipped solely with a video link, and the second with a pilot controller and data link.

  - – For my order, can you make one drone with a pilot controller, and another with just a video link? Thanks!

  - – Assemble: Drone 1: data link only. Drone 2: All possible modules

- Instructions with 3 drones

  - – Assemble three drones: the first should include a pilot controller, the second should have both a video link and a data link, and the third should have no extra modules.

  - – Build three drones. Give all modules to the first, only the pilot controller to the second, and leave the third with no extras.

  - – Could you please put data links in two drones and all the modules in the last one?

- Instructions with 5 drones

  - – Assemble 5 drones with a data link. The last one should have an extra pilot controller.

  - – Assemble 5 drones. The first drone should have all possible modules, the second and third drones should have only pilot controllers, the fourth drone should have a video link, and the fifth drone should have no extra modules.

  - – Assemble 5 drones. Equip the first drone with a data link and a video link, the second drone with a pilot controller and a video link, the third with only a data link, the fourth with only a pilot controller, and the fifth with no extra modules.

Table 3: Reasoning Ability Experiment Results.

| Tasks | RoBridge | CortexVLA |
|---|---|---|
| **Ambiguous Instructions** | 83.33% | 91.67% |
| **Execution Rejection** | 13.04% | 100.0% |

# F   EXTRA EXPERIMENTS

## F.1   REASONING ABILITY EVALUATION

We further evaluate the reasoning ability of CortexVLA in two different settings.

**Ambiguous Instructions**   Instead of clearly specifying the object to grasp, we provide the VLA with an ambiguous command requiring inference and reasoning based on context. The goal is to evaluate whether the agent is capable of using its intrinsic cognitive mechanisms to deduce the correct object to grasp. For example, we might instruct the model by saying, "I'm thirsty, get me something to drink, and expect the model to grasp a bottle of water and release it to the basket.

**Execution Rejection**   In this scenario, the VLA is provided with unreasonable or potentially dangerous instructions and is expected to reject execution. Moreover, it should ideally communicate the rationale behind its rejection to the user if capable of doing so. For example, the agent might be given the instruction, "Fetch the toy on the table," when, in fact, no toy is present on the table. In this case, the model should decline to execute the command and explain the reason for rejection.

The practical significance of this experiment is evident. In real-world applications, a reliable embodied intelligence should not behave as a machine that mechanically follows instructions without critical thought. Instead, it should possess a certain level of reasoning capability, enabling it to interpret ambiguous commands and, more importantly, to firmly and clearly refuse malicious or unreasonable requests. Such capabilities are essential not only for preventing user frustration but also for potentially avoiding catastrophic consequences.

The experimental results are presented in Table 3. RoBridge can interpret ambiguous instructions, but it struggles to perform execution rejection for lack of a safety or security configuration mechanism. Since end-to-end VLAs cannot understand instructions that differ from those used in fine-tuning, we exclude them from this experiment. These results demonstrate the CortexVLA's markedly stronger reasoning and adaptability. These findings expose a central weakness of the prevailing VLA paradigm. Despite their impressive execution abilities, their underlying cognitive competence remains shallow and insufficient for safe, dependable real-world use.

## F.2   ABLATION STUDY

The validation of Cortex-PPO with single SFT fine-tuning is presented in Table 1. Here, we further examine the effect of the noise injection component. To this end, we design a planning simulation using CortexVLA models trained with different values of $\sigma$. The simulation follows a setup similar to the main experiment but introduces greater variability in target objects. We also set the function success rate of one of the functions to 0.7, meaning that that function call has a 30% chance of random failure. The results, shown in Table 4, indicate no significant differences in planning success rates across different $\sigma$ values. This suggests that Cortex-PPO is robust to the choice of noise injection hyperparameter.

Table 4: Ablation of the noise distribution.

| Noise Distribution | Planning Success Rate |
|---|---|
| $\mu = 0, \sigma = 0.0095$ | 92.22% |
| $\mu = 0, \sigma = 0.0100$ | 90.00% |
| $\mu = 0, \sigma = 0.0105$ | 90.00% |

## G    LLM USAGE STATEMENT

We used LLMs solely to polish the writing of this paper.

