# OpenReview forum: "CortexVLA: Bridging the Gap between Cognition and Action via Function Calling"
_ICLR.cc/2026/Conference — Submitted to ICLR 2026_

### Official Review · Reviewer_BQ8W · 2025-10-25

**Soundness:** 3
**Presentation:** 3
**Contribution:** 3
**Rating:** 4
**Confidence:** 4

**Summary:**

This paper proposes CortexVLA, a modular framework designed to bridge cognition and action in robotic manipulation via LLM-based function calling. It consists of three components—the Central, Visual, and Motor “Cortices”—that handle planning, perception, and control respectively. The authors further introduce Cortex-PPO, a reinforcement learning algorithm that improves function-call robustness and enables failure recovery. The paper provides theoretical analyses of the algorithm’s unbiasedness and generalization, and reports strong experimental results on ultra-long-horizon and multi-scenario tasks, showing higher success rates and scalability compared to existing VLA and hierarchical baselines

**Strengths:**

- System integration and clarity: The paper presents a clean, modular architecture that unifies planning, perception, and control using LLM function calling. The system diagram and explanations are clear and easy to follow.
- Reproducibility: The methodology and baselines are well documented, and the authors commit to releasing code.
- Empirical performance: Extensive experiments across long-horizon tasks demonstrate consistent improvement over end-to-end and hierarchical VLA baselines. The evaluation is thorough and includes multi-scenario case studies (e.g., manufacturing, bartender tasks).

**Weaknesses:**

- Conceptual originality: While the system is presented under the “VLA” terminology, CortexVLA is more accurately a system-level orchestration framework that leverages existing LLM tool-calling capabilities, rather than a fundamentally new vision-language-action learning model. Its contribution lies primarily in engineering design and modular integration, rather than new algorithmic or scientific insights.
- Algorithmic contribution: Cortex-PPO appears to be a modest extension of standard PPO with reward shaping and noise injection. Theoretical analysis is mathematically sound but adds limited novelty or practical insight beyond existing RL formulations.
- Scope of evaluation: The experiments convincingly demonstrate performance, but it remains unclear how much of the gain stems from better task decomposition and function orchestration versus true advances in policy learning. The framework’s dependence on predefined tools may limit generalization in open-world settings.
- Positioning: The framing of CortexVLA as a “VLA model” might be misleading, as it does not train a unified vision-language-action policy. The work would be more accurately characterized as a tool-based orchestration framework for robot control.

**Questions:**

- How critical is Cortex-PPO to the reported performance improvements? A clear ablation isolating its contribution would strengthen the claim.
- How scalable is the system to tasks requiring unseen tools or novel function compositions?

---

> ### Author Response · Authors · 2025-11-20
> **Response to Reviewer BQ8W (Part 1/2)**
>
> **Dear Reviewer BQ8W,**
>
> Thank you for your valuable feedback. We appreciate your recognition of the model's performance and provide detailed clarifications below regarding the weaknesses and questions you raised.
>
> ---
>
> ### **1. Clarification of conceptual contribution and positioning of CortexVLA**
>
> We note that your first and fourth weaknesses share similar concerns, so we address them together. We clarify that the usage of "VLA" terminology is reasonable, and the contributions of our work include innovations in model frameworks, algorithms, and theories, not only engineering design and modular integration.
>
> We adopt the "VLA" terminology because CortexVLA indeed integrates Vision, Language, and Action modules, as it receives visual and linguistic inputs and generates actions. Regarding your concern about a "unified vision-language-action policy", we have discussed it in *Section 2.2* and *Appendix C* of our manuscript, including end-to-end training methods, formulations, and results. These demonstrate that our CortexVLA model supports end-to-end training.
>
> As stated in *Section 1* of our manuscript, the main conceptual contribution is proposing a novel Vision-Language-Action paradigm that enables reliable executions of ultra-long-horizon tasks. This is the area that existing VLA models often struggle with. Within this paradigm, we propose Cortex-PPO, an algorithm tailored to the structure of CortexVLA. Its effectiveness is demonstrated both theoretically and empirically, including the ablation presented in *Table 1* in our manuscript comparing **CortexVLA** and **CortexVLA (w/o Cortex-PPO)**.
>
> ---
>
> ### **2. Clarification of the novelty of Cortex-PPO**
>
> We clarify that Cortex-PPO is not a modest extension of standard PPO with reward shaping or noise injection. Its key contribution lies in introducing a failure-recovery-aware mechanism tailored for the function-calling process within the CortexVLA framework. The failure-recovery reward $\psi$ provides explicit learning signals that distinguish normal actions from recovery behaviors, which standard PPO does not capture, and Cortex-PPO further handles updates involving recovery differently from standard steps, which do not appear in prior PPO variants. In addition, our theoretical results, including unbiasedness of the noise-injected policy-gradient signal and an information-theoretic bound on cross-environment generalization, extend beyond standard PPO analyses. The experimental results further demonstrate the necessity and effectiveness of Cortex-PPO.
>
> ---
>
> ### **3. Clarification of performance gain and generalization in open-world settings**
>
> The third weakness you raised touches on two distinct concerns. We clarify them separately below.
>
> - ***Source of performance improvement:*** CortexVLA's strong performance arises from both (i) improved task decomposition and function orchestration enabled by the Central Cortex and (ii) policy learning enhanced by Cortex-PPO. The algorithm is explicitly designed to support the proposed function-calling framework and long-horizon decomposition in CortexVLA, enabling more reliable execution.
>
> - ***Generalization ability in open-world settings:*** In Section 3.2.1 (Scenario Adaptation -- Flexible Manufacturing), the model is applied *without further fine-tuning* and still achieves high success rates, indicating strong generalization to new environments. Additionally, we provide more evidence shown in the response to your second question (see the next comment block) to further validate the CortexVLA's generalization capabilities. Notably, *Supplementary Experiment 3* evaluates CortexVLA with previously unseen tools in a different usage context. The findings show that CortexVLA generalizes effectively even under tool distribution shifts, which highlights its potential for generalizing to open-world settings.
>
> Meanwhile, we emphasize that achieving full open-world generalization remains a major challenge for the existing VLA models [1], which still rely on a large amount of data collection and fine-tuning to successfully execute tasks with limited generalization capabilities. This is also a long-term goal for the robotics research community to achieve. While this broader challenge is beyond the scope of our manuscript, we view it as an important direction for future research. Thank you again for your valuable comments.
>
> ---
>
> ### **4. Clarification of the ablation of Cortex-PPO**
>
> As shown in *Table 1* of the manuscript, we have already provided the experiment results of **CortexVLA** and **CortexVLA (w/o Cortex-PPO)**, which can be considered as the ablation. This comparison of results demonstrates that the proposed Cortex-PPO algorithm is critical for the performance improvements.
>
> ---
>
> ### **References**
>
> [1] Fei, Senyu, et al. "LIBERO-Plus: In-depth Robustness Analysis of Vision-Language-Action Models." arXiv preprint arXiv:2510.13626 (2025).

---

> ### Author Response · Authors · 2025-11-20
> **Response to Reviewer BQ8W (Part 2/2)**
>
> ### **5. Response to the question about the scalability of the system**
> > **Corresponding question:** How scalable is the system to tasks requiring unseen tools or novel function compositions?
>
> To further evaluate the scalability of CortexVLA to tasks requiring unseen tools or novel function compositions, we conducted three supplementary experiments. The experimental details, results, and analysis are presented below.
>
> ---
>
> ### ***Supplementary Experiment 1: Scalability to unseen tools***
>
> To evaluate the scalability in tasks requiring unseen tools, we conducted a supplementary experiment based on the main experiment scenario. We introduce new tools that are not included in the training data, {`find`, `pick`, `place`, `retract`}, with new descriptions and arguments. Then, we conducted the experiments under the same settings as the main experiment. The results are shown in *Table 1* below. The results show that our model exhibits no significant performance degradation when encountering unseen tools, proving that our model possesses strong scalability to unseen tools.
>
> ***Table 1.  Experimental results with unseen tools and comparison with the original results.***
> | Sub-tasks Number | Success Rate | Original | Average Sub-task Length | Original |
> |------------------|--------------|----------|-----------------------|----------|
> | **8**  | **86.67%** | 84.38% | 7.00 | **7.41** |
> | **10** | **80.00%** | 80.65% | 8.60 | **9.58** |
> | **14** | 71.43% | **72.73%** | **12.29** | 11.55 |
> |
>
> ---
>
> ### ***Supplementary Experiment 2: Scalability to unseen tools and novel function compositions***
>
> To evaluate the scalability of CortexVLA on both unseen tools and novel function compositions, we conducted a supplementary experiment based on the Bartender case study. We add unseen tools, {`add_gin`, `add_ginger_beer`, `add_brandy`}, and novel function compositions, *Mojito (Gin + Ginger Beer)*, *Martini (Rum + Gin + Ginger Beer + Brandy)*. Without any fine-tuning, we evaluated the model's performance on these two novel function compositions using instructions, *"Give me a cup of Martini with ice"* and *"Mojito, please"*. We achieved a success rate of **90.91%**, showing no significant decline compared to the original success rate of 91.67%. This demonstrates our method's high scalability to unseen tools and novel function compositions.
>
> ---
>
> ### ***Supplementary Experiment 3: Scalability to unseen tools in a new usage context***
>
> To further evaluate the scalability of CortexVLA, we considered a task that requires the model to use unseen tools in a new usage context. Specifically, based on the unseen tools and novel function compositions we introduced in *Supplementary Experiment 2*, we add a `clean_table` function, which is fully different from the cocktail-making context that the model was fine-tuned from.  We used the instruction *"Make me a cup of Martini, and then clean the table"* to evaluate whether the CortexVLA could call functions in the correct order. The test results showed a **82.61%** success rate for function calls, indicating no significant drop in success rate. This further demonstrates the strong scalability of our model.
>
> ---
>
> In conclusion, across all three supplementary experiments, CortexVLA demonstrates strong scalability to unseen tools and novel function compositions.

---

### Official Review · Reviewer_eEQV · 2025-10-28

**Soundness:** 3
**Presentation:** 3
**Contribution:** 3
**Rating:** 8
**Confidence:** 3

**Summary:**

This paper proposes CortexVLA, a modular vision-language-action framework that bridges cognition and action via LLM-based function calling across three components: a Central Cortex for planning and orchestration, a Visual Cortex for perception, and a Motor Cortex for control. To enhance robustness on long-horizon tasks, the authors introduce Cortex-PPO, a recovery-aware PPO variant with noise injection, and provide theoretical analyses proving unbiasedness of the learning signal and an information-theoretic generalization bound. Experiments on ultra-long-horizon manipulation show strong performance and scalability, with an average success rate of 85.40% and sustained success on sequences up to 14 sub-tasks, outperforming end-to-end and hierarchical baselines. Case studies in flexible manufacturing (31 sub-tasks) and a bartender scenario further demonstrate adaptability, tool modularity, and failure recovery capabilities.

**Strengths:**

- Originality: The paper presents a distinctive modular VLA paradigm that leverages LLM function calling to explicitly bridge cognition (planning/state tracking) and action (perception/control) via the Central/Visual/Motor Cortices and a unified tool library. This design, coupled with failure-aware orchestration and persistent task-state handling, offers a novel alternative to end-to-end or purely hierarchical VLAs for long-horizon manipulation.

- Quality: The technical contribution of Cortex-PPO is well-motivated and supported by theory and practice. The empirical evaluation is thorough, spanning ultra-long-horizon benchmarks, ablations, and scenario adaptations (manufacturing and bartending), with clear performance gains over strong baselines.

- Clarity and Significance: The framework is clearly described with a concrete architecture, tool interfaces, and prompting templates, aiding reproducibility and deployment.

**Weaknesses:**

- The Central Cortex requires task-dependent prompts.
- End-to-end RL does not fine-tune the Visual Cortex. Gradients propagate only to the Motor Cortex and Central Cortex.

**Questions:**

- What are the concrete, real-world execution protocols for Cortex-PPO (e.g., safety gating, on-robot data collection pipelines, reset strategies, and hardware wear management)?
- How many environment interactions (samples) are required for real-world RL with Cortex-PPO?
- What are the components of Cortex-PPO’s reward function, and how are they evaluated in real-world settings?
- How can we systematically generate function-call training data for supervised fine-tuning (SFT)?
- For the evaluated tasks, how is the Motor Cortex fine-tuned?

---

> ### Author Response · Authors · 2025-11-20
> **Response to Reviewer eEQV (Part 1/2)**
>
> **Dear Reviewer eEQV,**
>
> Thank you for your thoughtful and constructive comments. We appreciate your recognition of the core innovations and contributions of our work. Below, we provide detailed explanations for the concerns you raised.
>
> ---
>
> ### **1. Response to the concern regarding "task-dependent prompts".**
>
> We understand your concern and provide additional clarification. In our framework, we categorize human-written prompts into two types.
> - ***System prompt:*** The system prompt guides the behavior and structure of the LLM, and is typically fixed for a certain scenario. As shown in Figures 5 and 6 of Appendix D, the *Planner*'s system prompt is task-dependent because it serves to inform the LLM of the task's fundamental details to enable accurate planning, which is necessary. This is conceptually similar to VLA fine-tuning, where a pretrained model is adapted to a specific task environment. In the *Executor*'s system prompt, only the mandatory tool description section is task-dependent, while the rest follows a universal format.
> - ***User prompt (task instructions):*** The user prompt plays the same role as natural-language instructions in end-to-end VLAs, specifying **what** the robot should do. For different tasks, the instructions will certainly be different.
>
> We agree that reducing or eliminating task-dependent prompts is an important research direction. However, given the current progress in the VLA field, achieving such strong generalization capabilities will require significant effort. We appreciate your valuable feedback and will continue to enhance our model's generalization performance in future work.
>
> ---
>
> ### **2. Response to the concern regarding "end-to-end RL does not fine-tune the Visual Cortex".**
>
> We apologize for the confusion caused by not providing a clearer explanation of this issue in the appendix, and we have updated this in the revision. The division of the Visual Cortex and Motor Cortex is not that distinct. For example, the model $\pi_0$ used in our implementation of the end-to-end RL method can be considered as both the Visual Cortex and Motor Cortex. As is standard in existing VLA works [1][2], we fine-tune only the action expert components, which are related to the Motor Cortex. Unfreezing the VLM (Visual-Cortex-related) for RL often destabilizes training and is not the typical VLA-finetuning protocol. Therefore, our end-to-end RL pipeline is theoretically capable of fine-tuning the Central, Visual, and Motor Cortex if desired.
>
> ---
>
> ### **3. Response to the question on generating function-call SFT data**
>
> To construct function-calling SFT data, we recommend building structured templates. The templates of the *system prompts* are shown in Figure 6 of Appendix D. The *user prompts* are used to describe specific tasks. We recommend leveraging powerful LLMs such as GPT to assist in generating diverse task instructions. The *assistant prompts*, which serve as the label of SFT, are the function calling contents that you want the LLMs to generate. We recommend constructing the function-calling contents in the predefined JSON format, which you can refer to the [Qwen Documentation](https://qwen.readthedocs.io/en/latest/framework/function_call.html). Finally, you need to add the function responses following the assistant prompt. For multi-round function-calling, as used in our work, we recommend aggregating these prompts in the following JSON structure:
>
> ```json
> [
>     [
>         {"role": "system", "content": SYSTEM_PROMPT},
>         {"role": "user", "content": USER_PROMPT},
>         {"role": "assistant", "content": "", "function_call": FUNCTION_CALL},
>         {"role": "function", "name": FUNCTION_NAME, "content": FUNCTION_RESPONSE},
>         {"role": "assistant", "content": "", "function_call": FUNCTION_CALL},
>         {"role": "function", "name": FUNCTION_NAME, "content": FUNCTION_RESPONSE},
>         ...
>     ]
> ]
> ```
>
> These data can be efficiently produced with Python scripts. We will also release our training data and further details after the manuscript is accepted.
>
> ---
>
> ### **4. Response to the question regarding fine-tuning of the Motor Cortex**
>
> In many scenarios, fine-tuning the Motor Cortex is unnecessary. For example, in the main experiments and the *Flexible Manufacturing* case study, we directly use AnyGrasp as the Motor Cortex, which does not need fine-tuning. When using ACT or VLAs as the Motor Cortex, as in the *Bartender* case study, we fine-tune the Motor Cortex following their original fine-tuning protocols.
>
> ---
>
> ### **References**
>
> [1] Black, Kevin, et al. "$\pi_0$: A Vision-Language-Action Flow Model for General Robot Control." arXiv preprint arXiv:2410.24164 (2024).
>
> [2] Kim, Moo Jin, et al. "Openvla: An open-source vision-language-action model." arXiv preprint arXiv:2406.09246 (2024).

---

### Official Review · Reviewer_28at · 2025-10-31

**Soundness:** 2
**Presentation:** 3
**Contribution:** 3
**Rating:** 4
**Confidence:** 3

**Summary:**

The paper introduces CortexVLA, a modular Vision¨CLanguage¨CAction (VLA) architecture that leverages LLM-based function calling to bridge high-level cognition and low-level control. The system separates planning (Central Cortex), perception (Visual Cortex), and action execution (Motor Cortex), with a dedicated task-handler for maintaining task state and managing bounded prompts. To enhance robustness and recovery from execution errors, the authors further propose Cortex-PPO, a PPO-based finetuning algorithm that incorporates a recovery-aware bounded reward (Cortex-Reward) and additive Gaussian noise. Theoretical analysis is provided to show that the noise-injected reward remains an unbiased estimator for policy gradients. Empirically, CortexVLA achieves higher success rates on ultra-long-horizon manipulation tasks and a flexible manufacturing scenario compared with several end-to-end and planning-based baselines.

**Strengths:**

1. The modular VLA architecture is well-motivated and clearly described, with distinct components for perception, planning, and control, as well as a task-handler for state tracking. Long-horizon error accumulation is a concrete challenge for current VLAs; decoupling cognition, perception, and action via callable tools is a natural and promising direction to train in end-to-end way.
2. The introduction of Cortex-PPO addresses a practical weakness of supervised fine-tuned VLAs: their inability to recover after partial execution errors. The recovery-aware bounded reward and the theoretical justification for the noise injection strategy are well presented and contribute novel insights to improving stability and generalization.

**Weaknesses:**

1. The main experiments rely almost exclusively on repetitive pick-and-place tasks, differing only by object identity or order. Even the so-called ¡°ultra-long-horizon¡± tasks (e.g., those with 14 subtasks) are essentially the same operation repeated multiple times. This design inflates the horizon length without increasing task diversity or reasoning complexity. To convincingly demonstrate the benefits of CortexVLA, the paper should include a broader set of tasks that require different skills, reasoning steps, or compositional dependencies.

Therefore, in this kind of repetitive tasks, it is no surprise that end-to-end methods will fail in the long-horizon setting. Moreover, the hierarchical baseline is not described in enough detail to understand why it fails;
With a well-tuned hierarchical method, these tasks should, in principle, be solvable at least in the planning stage. As such, the current experiments are not strong enough to substantiate the claimed advantages of CortexVLA, and the single task family limits the generality of the conclusions.


2. For the Case Studies part, the paper propose two kinds of scenarios: flexible manufacturing and cocktail making. The results show that CortexVLA performs well, but there is no comparison with any baseline methods. It is hard to judge how much improvement CortexVLA brings without any baselines. Including even a simplified baseline (e.g., an end-to-end VLA or hierarchical planner) would make these results more credible.

**Questions:**

The most significant questions are already listed under weaknesses. In addition, I would be interested to get a clarification on the following point:
+ Line 1099: The paper claim that CortexVLA substantially outperforms all baseline methods as shown in Table 3. But Table 3 does not appear to include any baseline results.

---

> ### Author Response · Authors · 2025-11-21
> **Response to Reviewer 28at (Part 1/2)**
>
> **Dear Reviewer 28at,**
>
> Thank you for your valuable and constructive feedback. We appreciate your recognition of the proposed model architecture and algorithm. We provide detailed clarification below regarding the weaknesses and questions you raised.
>
> ---
>
> ### **1. Response to Weakness 1**
>
> Your first point raises concerns about the strength of the experimental setup and the reasons behind the failure of the hierarchical baselines. We clarify both points below.
>
> ---
>
> > ***1.1 Clarification on the experimental design***
>
> We believe our experimental design considers the task diversity and reasoning complexity. The experiments reported in the manuscript include not only tasks requiring pick-and-place operations but also the *Bartender* task described in *Section 3.2.2*, which requires dexterous operations. This task involves not only pouring liquids but also picking ice cubes and placing them into cups, reflecting **task diversity**. Additionally, the model must correctly understand user requests, which refers to **reasoning complexity**. As shown in our results, CortexVLA achieves high success rates across all three evaluated task scenarios.
>
> Moreover, long-horizon tasks commonly studied in prior work (e.g., desk sorting or cleanup [1]) also consist of multiple pick-and-place operations on different objects. Increasing the number of sub-tasks is a standard approach to raising task difficulty and evaluating scalability. Our task design, therefore, not only reflects general manipulation tasks but also extends to more challenging, assembly-like settings. This is a key motivation for establishing the *Flexible Manufacturing* task in *Section 3.2.1*. Therefore, we believe our experimental design is capable of demonstrating the capabilities of CortexVLA.
>
> ---
>
> > ***1.2 Explanation on the failure of hierarchical baselines***
>
> We first elaborate on some concepts and then provide detailed analyses for the failure of hierarchical baselines.
>
> First, we clarify that the "failed" mark shown in *Table 1* of the manuscript indicates that the method could not complete the entire long-horizon task, though its average success length (as shown in *Figure 2*) may still not be short, for example, RoBridge. This means the baseline makes progress but consistently fails at some step in long-range execution.
>
> Second, the "planning" stage in hierarchical methods is not limited to scheduling the order of sub-tasks; it also requires determining whether each sub-task has been completed. For example, RoBridge breaks down each sub-task into detailed actions, so that the model must confirm whether each action has been completed after execution. We acknowledge that the original manuscript may not have sufficiently emphasized this point and have clarified it in the revision.
>
> Below, we provide further explanation of the reasons for the failure of the hierarchical baselines.
>
> - ***RoBridge:*** The internal implementation of RoBridge is closed-source, which relies on GPT-4o to perform the planning. Therefore, we cannot perform fine-tuning. In our experiments, we assume that the actions output by RoBridge can successfully grasp objects, and we provide images of each completed step as reported in the RoBridge paper, which gives it extra advantages. However, the most common failure of RoBridge is misjudging task progress, particularly misdetecting whether the object has been successfully grasped by the gripper.
>
> - ***Hierarchical VLA-OS (VLA-OS-H):*** It generates task plan tokens through its VLM layer. By decoding these plan tokens, we can evaluate the maximum possible success rates, which also gives it extra advantages. But we still observed limited performance. Common errors for VLA-OS-H include planning an extra object to operate on or assigning an incorrect object name to a sub-task.
>
> ---
>
> In summary, the experimental results across multiple scenarios highlight the strong performance and robustness of CortexVLA. We appreciate your comments on potential limitations. We will explore the possibility of implementing more diverse task combinations in future work.
>
> ---
>
> ### **References**
>
> [1] Fan, Yiguo, et al. "Long-vla: Unleashing long-horizon capability of vision language action model for robot manipulation." arXiv preprint arXiv:2508.19958 (2025).

---

> ### Author Response · Authors · 2025-11-21
> **Response to Reviewer 28at (Part 2/2)**
>
> ### **2. The performance of baseline methods in the Case Studies**
>
> We apologize for any confusion caused by not reporting the baseline performance in the case studies. The omission was because case studies are substantially more challenging for the baseline methods. For example, in *Flexible Manufacturing*, assembling a single drone already constitutes a 6-subtask long-horizon task, which end-to-end methods were unable to complete even in the main experiment. An end-to-end VLA may be able to handle a simple deterministic sub-task, such as grasping a part of the drone, but it cannot complete the entire long-horizon task since the task length is too long. Hierarchical baselines face similar limitations as described earlier in point 1.2 in the above comment block.
>
> Nevertheless, for completeness, we provide supplementary results for the baseline methods in *Table 1* and *Table 2* below.
>
> ***Table 1. Supplementary results of baseline methods in the Flexible Manufacturing case study.***
> | Number of Drones | Avg. Succ. Length (RoBridge) | Avg. Succ. Length (CortexVLA) | Succ. Rate (RoBridge) | Success Rate (CortexVLA) |
> |---|---|---|---|---|
> | 1 drone | 3.38 | **6.00** | 23.08% | **100.00%** |
> | 2 drones | 5.29 | **12.94** | failed | **94.12%** |
> | 3 drones | 7.11 | **17.30** | failed | **86.96%** |
> | 5 drones | 9.08 | **26.69** | failed | **81.25%** |
> |
>
> ***Table 2. Supplementary results of baseline methods in the Bartender case study.***
>
> | Method | Success Rate |
> |---|--------------|
> | $\pi_0$ | 9.52% |
> | RoBridge | 21.43% |
> | **CortexVLA** | **91.67%** |
> |
>
> ---
>
> ### **3. The performance of baseline methods in ambiguous instructions and execution rejection**
>
> We apologize for not updating the results earlier in the appendix. We have included the new results in the revision and presented them in *Table 3* below. Since end-to-end baselines, such as $\pi_0$, cannot understand instructions that differ from those used in fine-tuning, we exclude them from this experiment. RoBridge’s performance on execution rejection is limited by its lack of a safety or security configuration mechanism.
>
> ***Table 3. Results of ambiguous instructions and execution rejection.***
> | Tasks | RoBridge | CortexVLA |
> |---|---|---|
> | Ambiguous Instructions | 83.33% | **91.67%** |
> | Execution Rejection | 13.04% | **100.00%** |
> |

---

### Official Review · Reviewer_eVQ9 · 2025-11-01

**Soundness:** 3
**Presentation:** 2
**Contribution:** 2
**Rating:** 2
**Confidence:** 4

**Summary:**

This paper proposed to address several challenges arising from the long-horizon tasks in robotic tasks, via modularized model from a three-cortex architecture: Central Cortex (LLM-based cognition hub), Visual Cortex (perception toolset), and Motor Cortex (action controller). The Central Cortex is designed with a planner and a task handler for persistent sub-task state; the visual cortex is designed with callable visual models like grounding DINo, SAM + D3RoMa for visual information; and the motor cortex designed with pose predictor or motion generators to generate motions. These 3 cortices are linked via function calls and trained with a novel Cortex-PPO reinforcement learning algorithm, with a recovery-aware reward and additive noise injection. CortexVLA reports 85.40% average success rate and 72.73% success at 14 steps with avg. sub-task length 11.55, outperforming end-to-end and hierarchical baselines.

**Strengths:**

* Architectural clarity: there is a clean separation of cognition (planning + allocation), perception, and control, with a unified tool library and function descriptions that encode recovery strategies. This directly targets error propagation and context bloat in long sequences.
* Design of recovery-aware RL: the authors propose Cortex-reward thay can stabilize training, encourage retries, and mitigate reward sparsity, with a theoretically clean unbiasedness argument for GAE-PPO under additive noise and an information-theoretic rationale for better cross-environment generalization, providing a solid theoretical bound for the method.
* Long-horizon capability: task handler explicitly tracks sub-task progress to prevent loss in the long context; the results are also promising.

**Weaknesses:**

* Computational cost could be high: as the system consists of several external tools, like for the visual cortex, it uses GroundingDINO and other pretrained vision models, which could cause a high computational burden of the whole method, also cause high latency for function-calling and execution of the external tools. Seems there is no timing, token budget, or cycle-time analysis; this matters for real-time manipulation and scalability to denser action frequencies.
* Predefined tool schemas make it less robust and adaptive: I notice that the system is dependent on well-written tool descriptions and explicit recovery strategies in function specs. This might cause the method to be prompt-sensitive and brittle across new tool APIs without careful prompting.
* Evaluation metrics are insufficient & lack of ablation: the authors claimed their method is capable of recovering from the errors; however, there is no corresponding evaluation metric designed or used to evaluate this capability, or qualitative results to showcase the effectiveness, instead, there is a heavy emphasis on success rate and avg. success length.

**Questions:**

* Can the authors provide more experimental results of the method on other commonly used evaluation benchmarks? Or using more metrics like per-subtask accuracy, number of recoveries invoked, action efficiency (steps/time), and compounding error curves by category would be helpful as well. These could provide more context and insights into your method's capability.
* If there are new tool schemas to be added to the framework, how can this be done?
* How does the model's generalization capability improve by introducing the modular components? Are they becoming more robust?

---

> ### Author Response · Authors · 2025-11-21
> **Response to Reviewer eVQ9 (Part 1/3)**
>
> **Dear Reviewer eVQ9,**
>
> Thank you for your thoughtful and constructive comments. We appreciate your recognition of the proposed model architecture and algorithm design. We provide detailed clarification below regarding the weaknesses and questions you raised.
>
> ---
>
> ### **1. Response to the concern on the computational cost**
>
> First, we clarify that the use of external tools does not introduce substantial computational overhead and is not the bottleneck for real-time manipulation or tasks requiring denser actions. The external tools we use, such as Grounding DINO, contain only 172M parameters, far smaller than typical VLMs or LLMs, making their cost negligible in comparison. Since our framework is designed to flexibly integrate external tools, faster tools can be substituted when tasks require higher-frequency control. For example, as discussed in *Section 3.2.2 Scenario Adaptation -- Bartender*, we adopt ACT (80M) precisely to keep computational cost low and ensure efficient execution.
>
> We also conducted a FLOPs comparison. With an image resolution of 224×224, an inference frequency of 20Hz, and an action chunk length of 50, $\pi_0$ performs roughly $2.0 \times 10^{13}$ FLOPs to complete one sub-task (around 10 seconds). CortexVLA requires approximately $1.8 \times 10^{13}$ FLOPs to process 1300 tokens and complete a sub-task. This shows that our computational cost is on par with mainstream VLAs, while offering greater flexibility and adaptability. Furthermore, cost can be further reduced through software–hardware co-design, such as parallelization and tool preloading. Besides, given the higher robustness of our method, it is worthwhile for the time expense.
>
> Second, we clarify that the function calling latency is not high. Although an LLM may incur some "thinking" cost before doing a function call, the Central Cortex trained with Cortex-PPO has learned to trigger the think behavior selectively. In practice, thinking is used mainly for failure recovery, while regular function calls do not trigger it. Also, with KV caching, standard functions execute quickly. For reference,  as reported in the $\pi_0$ paper [1], the inference latency of a 20Hz robot is **0.8s**. For CortexVLA, under the experimental setting with 8 sub-tasks, the average latency of function calls is **0.510 ± 0.014** (s) without failure recovery and **0.867 ± 0.042** (s) with failure recovery, which is comparable to $\pi_0$. Given the substantial gains in task success rate and average success length, this level of latency is negligible for most tasks.
>
> Although this manuscript focuses on enhancing ultra-long-horizon task performance, we fully acknowledge the importance of computational efficiency and will explore further cost reductions in future work.
>
> ---
>
> ### **2. Response to the question about adding new tool schemas**
> > **Corresponding question:** If there are new tool schemas to be added to the framework, how can this be done?
>
> Adding new tool schemas is straightforward. The essential step is to extend the system prompts following the templates provided in *Appendix D*, which specify how tool descriptions should be written. Fine-tuning is optional and depends on the model’s inherent understanding. If needed, one can construct corresponding data for SFT, followed by Cortex-PPO training. Full implementation details will be released upon acceptance.
>
> ---
>
> ### **3. Response to the question regarding generalization capability**
> > **Corresponding question:** How does the model's generalization capability improve by introducing the modular components? Are they becoming more robust?
>
> Modular components improve generalization by allowing each module to be selected from tools that already generalize well. For example, in the main experiment, the Visual Cortex (Grounding DINO) generalizes more effectively to novel object categories than VLM-based perception, and the Motor Cortex (AnyGrasp) predicts grasp poses for diverse unseen objects without retraining. By contrast, VLA models typically require new data collection and fine-tuning even for a single object. In addition, the Central Cortex (LLM) contributes substantial generalization due to its broad knowledge base, enabling it to understand diverse task instructions and tool usages across different domains.
>
> As evidence, in the *Flexible Manufacturing* case study, we directly transfer the CortexVLA model trained in the main experiment settings to this new scenario **without any fine-tuning**, and the model still maintains exceptionally high performance on ultra-long-horizon tasks. Existing VLA models require additional data collection and fine-tuning when transferred to new scenarios. This demonstrates superior generalization capability compared to mainstream VLAs, and also reflects the robustness improvements compared to baselines.
>
> ---
>
> ### **Reference**
>
> [1] Black, Kevin, et al. "$\pi_0$: A Vision-Language-Action Flow Model for General Robot Control." arXiv preprint arXiv:2410.24164 (2024).

---

> ### Author Response · Authors · 2025-11-21
> **Response to Reviewer eVQ9 (Part 2/3)**
>
> ### **4. Response to the concern on the predefined tool schemas**
>
> We clarify that the use of predefined tool schemas does not mean the method is prompt-sensitive, and does not make the model less robust or adaptive. We demonstrate this through analyses and experiments.
>
> For LLM-based tool calling, providing basic tool information in the prompt is a fundamental requirement [1][2]. Since tool interfaces are human-defined, the LLM must be given the tool names, arguments, descriptions, and recovery strategies; otherwise, it has no way of inferring how the tools should be invoked. However, these descriptions do not need to be well-written. Modifying the function's description in different ways does not affect the function calling accuracy in our framework.
>
> At the same time, the presence of predefined tool schemas does not reduce CortexVLA's robustness or adaptability. On the contrary, CortexVLA can flexibly incorporate new tool APIs and use them effectively without carefully engineered prompts or additional fine-tuning. To demonstrate this, we conducted three supplementary experiments.
>
> ---
>
> > **4.1 Supplementary Experiment 1: Adaptability to unseen tool APIs**
>
> To evaluate CortexVLA's adaptability to new tool APIs, we introduce new tool APIs, {`find`, `pick`, `place`, `retract`}, with new descriptions and arguments not included in training. As shown in *Table 1*, under the same experimental settings as the main experiment, CortexVLA shows no significant performance degradation. The results demonstrate that CortexVLA maintains strong performance even with additional unseen APIs, reflecting strong adaptability.
>
> ***Table 1.  Experimental results with new tool APIs vs. original results.***
> | Sub-tasks Number | Success Rate | Original | Average Sub-task Length | Original |
> |---|---|---|---|---|
> | **8**  | **86.67%** | 84.38% | 7.00 | **7.41** |
> | **10** | **80.00%** | 80.65% | 8.60 | **9.58** |
> | **14** | 71.43% | **72.73%** | **12.29** | 11.55 |
> |
>
> ---
>
> > **4.2 Supplementary Experiment 2: Robustness to new tool APIs and new compositions**
>
> To further evaluate the robustness and adaptability of CortexVLA, we conducted a supplementary experiment based on the *Bartender* case study. We introduced new tool APIs, {`add_gin`, `add_ginger_beer`, `add_brandy`}, and new cocktail recipes, *Mojito (Gin + Ginger Beer)*, *Martini (Rum + Gin + Ginger Beer + Brandy)*. Without any fine-tuning, CortexVLA successfully executed instructions such as *"Give me a cup of Martini with ice"* and *"Mojito, please"*, achieving a **90.91%** success rate, comparable to the original of 91.67%. This demonstrates strong adaptability and robustness when facing new tool APIs and new compositions.
>
> ---
>
> > **4.3 Supplementary Experiment 3: Adaptability to new tools APIs in an unseen usage context**
>
> To further evaluate the adaptability of CortexVLA, we considered a task that requires the model to use new tool APIs in an unseen usage context. Building on *4.2 Supplementary Experiment 2*, we added a `clean_table` tool, which lies completely outside the cocktail-making context used during fine-tuning. Using the instruction *"Make me a cup of Martini, and then clean the table"*, CortexVLA achieved an **82.61%** success rate for tool calls, again with no significant performance drop. This further demonstrates the strong adaptability to unseen scenarios of CortexVLA.
>
> ---
>
> In conclusion, through the three supplementary experiments, we strongly demonstrated that CortexVLA exhibits high robustness and adaptability for new tool APIs, and predefined tool schemas will not make it less robust and adaptive.
>
> ---
>
> ### **Reference**
> [1] Liu, Weiwen, et al. "Toolace: Winning the points of llm function calling." arXiv preprint arXiv:2409.00920 (2024).
>
> [2] Qin, Yujia, et al. "Toolllm: Facilitating large language models to master 16000+ real-world apis." arXiv preprint arXiv:2307.16789 (2023).

---

> ### Author Response · Authors · 2025-11-21
> **Response to Reviewer eVQ9 (Part 3/3)**
>
> ### **5. Response to the concern regarding the evaluation of failure recovery ability**
>
> We did not conduct a separate qualitative analysis of failure recovery capability in the manuscript because the failures in real-world environments are complex and unpredictable. For example, an unexpected hardware transmission error can cause an execution to fail. However, the average success rate we reported can indirectly reflect the model's failure recovery ability. This is because if the recovery capability were poor, the average success rate would not be so high. For this reason, we focus on success rate and average success length, which are the commonly used evaluation metrics in the relevant studies [1][2]. These two metrics provide a more objective and comprehensive measure of a model's capability in long-horizon tasks.
>
> To further demonstrate CortexVLA's failure recovery ability, we additionally conducted a supplementary experiment. Using the main experimental configuration and the first 8-subtask instruction (shown in *Appendix E.1* of the manuscript), we manually induced grasping failures at each sub-task by temporarily removing and replacing the object. We then recorded the model's recovery success rate at each step. As shown in *Table 2*, the results indicate that CortexVLA exhibits strong and consistent failure recovery capabilities across all sub-tasks.
>
> ***Table 2. Failure Recovery Success Rate Results***
> | Sub-task Index | 1 |  2 | 3 | 4 | 5 | 6 | 7 | 8 | Avg. Succ. Rate |
> |---|---|---|---|---|---|---|---|---|---|
> | **Failure Recovery Succ. Rate** | 92.31% | 100.00% | 84.62% | 100.00% | 69.23% | 100.00% | 92.31% | 84.62% | **90.38%** |
> |
>
> ---
>
> ### **6. Response to the question about more experimental results**
>
> In robotics research, real-world experiments are generally regarded as more meaningful than simulation-based evaluations, since simulations often fail to capture the complexity and variability of real-world conditions. Therefore, our paper primarily focuses on real-world experiments, and the corresponding experimental settings and implementation details will be released. We greatly appreciate your valuable suggestions. Our future work involves developing an online real-world benchmark, enabling users to remotely control robots for evaluation on our platform.
>
> Regarding additional metrics, we provide several supplementary results. *Table 3* reports the per-subtask accuracy across different task lengths. *Table 4* reports the average number of recoveries invoked and the three types of failures. Note that a failure does not necessarily lead to a failure recovery operation, as some failures may directly cause the entire task to fail. The action efficiency we calculated is 0.14 function calls per second and 0.04 sub-tasks per second. While the primary goal of our work is to demonstrate reliable execution over ultra-long horizons, we hope these supplementary results offer a more comprehensive view of the model's behavior.
>
> ***Table 3.  Per-subtask accuracy results.***
> | Number of Subtasks | 1 | 2 | 3 | 4 | 5 | 6 | 7 | 8 | 9 | 10 | 11 | 12 | 13 | 14 |
> |---|---|---|---|---|---|---|---|---|---|---|---|---|---|---|
> | **1** | 0.941 | - | - | - | - | - | - | - | - | - | - | - | - | - |
> | **2** | 1.000 | 0.909 | - | - | - | - | - | - | - | - | - | - | - | - |
> | **3** | 0.941 | 0.969 | 1.000 | - | - | - | - | - | - | - | - | - | - | - |
> | **4** | 0.973 | 0.917 | 0.970 | 1.000 | - | - | - | - | - | - | - | - | - | - |
> | **5** | 0.971 | 1.000 | 1.000 | 1.000 | 0.882 | - | - | - | - | - | - | - | - | - |
> | **8** | 1.000 | 1.000 | 1.000 | 0.969 | 0.903 | 1.000 | 0.964 | 1.000 | - | - | - | - | - | - |
> | **10** | 1.000 | 1.000 | 1.000 | 1.000 | 1.000 | 0.968 | 1.000 | 0.800 | 1.000 | 1.000 | - | - | - | - |
> | **14** | 1.000 | 1.000 | 1.000 | 1.000 | 1.000 | 0.727 | 1.000 | 1.000 | 1.000 | 1.000 | 1.000 | 1.000 | 1.000 | 1.000 |
> |
>
> ***Table 4.  Results of the average number of recoveries invoked and three types of failure.***
> | Number of Subtasks | 1	| 2 | 3 | 4 | 5 | 8 | 10 | 14 |
> |---|---|---|---|---|---|---|---|---|
> | **Avg. Num. of Recoveries Invoked** | 0.17 | 0.31 | 0.58 |	0.83 | 1.08 | 1.58 | 2.08 | 2.58 |
> | Avg. Num. of Grasp Failure |	0.17 | 0.23 | 0.42 | 0.83 | 0.92 | 1.42 | 1.75 | 2.25 |
> | Avg. Num. of Trajectory Planning Failure | 0.00 | 0.08 | 0.17 | 0.08 | 0.17 | 0.25 | 0.25 | 0.17 |
> | Avg. Num. of Hardware Transmission Failure | 0.00 | 0.15 | 0.08 | 0.17 | 0.17 | 0.08 | 0.23 | 0.42 |
> |
>
> ---
>
> ### **References**
>
> [1] Fan, Yiguo, et al. "Long-vla: Unleashing long-horizon capability of vision language action model for robot manipulation." arXiv preprint arXiv:2508.19958 (2025).
>
> [2] Zhang, Kaidong, et al. "RoBridge: A Hierarchical Architecture Bridging Cognition and Execution for General Robotic Manipulation." arXiv preprint arXiv:2505.01709 (2025).

---

### Author Response · Authors · 2025-11-24
**Response to All Reviewers**

We thank all reviewers for their careful reading and constructive feedback. We appreciate the recognition of our core contributions, including the modular architecture, the theoretically grounded algorithmic design, and the strong performance of CortexVLA on ultra-long-horizon real-world tasks. Below, we summarize the key strengths highlighted by the reviewers and provide an overview of how we have addressed the main questions and concerns.

#### ***Strengths highlighted by reviewers***

- **Clear and well-motivated modular architecture.** Reviewers consistently recognized the novelty and clarity of CortexVLA’s modular design, which effectively bridges cognition, perception, and control through unified tool interfaces and state tracking.
- **Novel recovery-aware RL algorithm.** Reviewers emphasized the novelty of the proposed Cortex-PPO algorithm across the recovery-aware reward design and solid theoretical justification.
- **Strong empirical performance.** Reviewers highlighted the consistent improvements over end-to-end VLAs and hierarchical baselines across diverse ultra-long-horizon tasks and real-world scenarios.

#### ***Supplementary clarification of our method***

- **Key contributions.** The significance of our framework lies in its function-based approach and modular design, enabling the selection of the most suitable module for different task scenarios and allowing each module to maximize its effectiveness. Our methodology has demonstrated an order-of-magnitude improvement in robustness compared to baseline methods during experimentation.
- **Error accumulation avoidance.** A common weakness of existing VLA approaches is severe error accumulation over long horizons. In contrast, CortexVLA exhibits minimal error propagation, owing to its closed-loop, autoregressive function-call chaining. Each sub-task is executed as an independent, self-contained module, with failures resolved within the sub-task through recovery mechanisms, preventing errors from cascading into subsequent steps.

#### ***Summary of key questions and our responses***

- **Adaptability and scalability to new tools and new function compositions.** Several reviewers asked about CortexVLA’s ability to handle new tools and novel function compositions. We conducted three additional experiments demonstrating its strong adaptability and consistent performance under unseen tools and compositions.
- **Evaluation metrics and results.** Following reviewers’ suggestions, we expanded our evaluation to include additional metrics such as failure recovery success rate, per-subtask accuracy, average number of recoveries, average number of failures by categories, action efficiency, and computational cost. We also provide supplementary results for baseline methods in the case studies to provide a more complete comparison.
- **Details of Cortex-PPO and end-to-end RL.** We clarified the contributions of Cortex-PPO, provided additional implementation details, and explained the optimization objectives for end-to-end RL.

We are grateful for the reviewers' thoughtful feedback, which has enabled us to substantially enhance the paper's integrity and emotional depth. We believe the revised version addresses all comments. Should any questions or uncertainties remain, we are happy to provide further clarification.

---

### Meta-Review · Area_Chair_nMZP · 2026-01-07

**Summary:**

Reviewers agree on the paper’s architectural clarity, the usefulness of modular design, and its empirical performance on ultra-long-horizon tasks compared with the baselines.
However, key concerns include:

- Practical issues: Computational cost, less robust and adaptive, (new tool with/without careful prompting.), simple action tasks (mostly pick-and-place).
- incomplete evaluations: missing baselines in some experiments; lack of failure recovery evaluations; lack of common benchmarks/metrics (e.g., efficiency and speed);
- positioning and novelty concerns: VLM + tool calling instead of VLA; unclear advantage of Cortex-PPO over standard RL baselines.

**Reviewer Concerns:**

Fully or partially addressed:
- Computational cost and latency: FLOPs and tool-calling latency are reported, but it remains unclear whether the latency matches the control frequency of compared baselines or on which hardware (e.g., GPU) measurements were taken.
- Zero-shot tool usage is shown. Task-specific prompting is required.
- Evaluations on failure recovery and analysis are provided.
- New results on ambiguous instructions and execution rejection are provided.
- Baseline results are provided in the case studies and extra experiments.
- End-to-end training of all three modules is explained conceptually but not empirically evaluated.

Still outstanding:
- No direct comparison of Cortex-PPO against standard RL baselines.
- Task diversity remains limited: most tasks are repetitive pick-and-place; the rebuttal defends increasing subtask count as standard.
- No evaluation on common benchmarks.
- Baseline fairness: Hierarchical baselines are not fine-tuned, and their components (e.g., LLM, policy) may be outdated compared to CortexVLA’s, making it unclear whether performance gains stem from architectural innovation or stronger individual modules.

**Reviewer Scores:**

eVQ9 may maintain or slightly improve the score. Many concerns including computational cost, latency, tool adaptability, failure recovery are answered, though some details are missing. Common benchmarks are still absent.

28at is likely to keep the score. Despite added baseline results and defense of task design, the central concern of lack of task diversity remains.

eEQV may lower the score. The two weaknesses are clarified but not fully resolved. Questions about some details of Cortex-PPO are not adequately answered.

BQ8W is likely to maintain the score. Some evaluations and clarifications of CortexVLA are provided, but the ablation for Cortex-PPO remains insufficient.

---

### Decision · Program_Chairs · 2026-01-26

Reject